# A systematic review and network meta-analysis of randomized controlled trials of well-being-focused interventions

Lowri Wilkie [1,2] ✉, Zoe Fisher[2,3,4], Antonia Geidel[1], Isabel Goodall[1], Shannon Kamil[1], Elen Davies[1] & Andrew Haddon Kemp [1,2] ✉

Improving population well-being is increasingly recognized as a global priority, yet evidence on the comparative effectiveness of well-being-focused interventions in adults is fragmented. Here we conduct a preregistered systematic review and network meta-analysis (PROSPERO CRD42023403480) of randomized controlled trials evaluating well-being interventions in adults without diagnosed conditions. Searches of MEDLINE, PsycINFO, CENTRAL and Scopus (to March 2023) identified 183 trials ($n$ = 22,811). Interventions included mindfulness-based, compassion-based, acceptance and commitment therapy and positive psychology interventions, as well as exercise, yoga, educational, nature-based programmes and combined exercise-psychological approaches. Risk of bias was assessed using RoB 2, and data were synthesized using random-effects network meta-analysis. Most interventions improved well-being compared with inactive controls. Combined exercise-psychological interventions produced the largest effect (standardized mean difference of 0.73, 95% confidence interval 0.27 to 1.20). Mindfulness, compassion, single positive psychology, yoga and exercise interventions demonstrated moderate, consistent effects (standardized mean difference of 0.41–0.49), with no significant differences between interventions. Nature-based interventions were not significantly more effective than controls, but evidence was limited by conceptual and methodological heterogeneity. Risk of bias was frequently moderate to high, and funnel plot asymmetry suggested potential publication bias. However, multiple sensitivity analyses (including grey literature, excluding studies with high risk of bias and small studies) supported the robustness of overall conclusions. Most comparisons (71%) were rated as moderate in certainty of evidence using CINEMA. These findings provide an integrated synthesis of the well-being intervention literature and highlight priority areas for future interdisciplinary, methodologically robust research. No external funding was received.

[1]School of Psychology, Faculty of Medicine, Health and Life Science, Swansea University, Swansea, UK. [2]Regional Neuropsychology and Community Brain Injury Service, Morriston Hospital, Swansea, UK. [3]Health and Wellbeing Academy, Faculty of Medicine, Health and Life Science, Swansea University, Swansea, UK. [4]Strategy Directorate, Swansea Bay University Health Board, Baglan HQ, UK. ✉e-mail: lowriwilkie@gmail.com; a.h.kemp@swansea.ac.uk

Global health is increasingly shaped by interrelated challenges including the rise of non-communicable diseases[1–3], widening health inequalities[4–6] and the accelerating impacts of climate change[7–9]. In response, there is growing recognition that enhancing well-being across individuals, communities, and ecosystems provides an integrative strategy for addressing pressing issues[10–12]. Traditionally, health interventions have targeted illness and dysfunction, yet, a growing body of research links the promotion of well-being to improved public health outcomes[13,14], greater resilience[15], prevention of mental disorders[16], stronger social bonds[17] and increased sustainable behaviours[18]. However, the relative effectiveness of different well-being-focused interventions across disciplines remains unclear.

Substantial research in the field of well-being has centred on psychological interventions explicitly designed to enhance well-being through positive psychology interventions (PPIs)[19–22]. However, despite evidence that factors such as exercise, nutrition and sleep substantially impact mental health[23–25], physical health-focused interventions are often studied separately from psychological well-being interventions. Well-being science has also faced criticism for focusing too closely on the individual, with less consideration for the social and environmental contexts in which the individual is embedded[26]. In response to these criticisms, a need arose for a framework to consolidate existing scholarly literature, multidisciplinary research and diverse theories into an integrated model. A framework was required to consider the interaction between mind and body, potential physiological underpinnings of well-being and broader collective and environmental context beyond the individual. Meeting this need, the GENIAL theoretical framework was developed to offer a structured approach to understanding well-being by integrating multiple disciplinary perspectives. The GENIAL model is a theoretical, interdisciplinary framework of the well-being literature[27–30]. The framework summarizes the key determinants of well-being relating to the individual (the importance of both mind and body connection including emotional regulation, sense of meaning and purpose and health behaviours), the community (including social connection, social cohesion and social capital) and the planet (connection to nature and sustainable practices). We have summarized well-being as a connection to the self, others and planet[31]. This holistic definition is supported by evidence demonstrating the efficacy of well-being-promoting interventions from across disciplinary domains including psychological interventions[19], exercise[32], social support[33] and nature connection[34].

Psychological well-being interventions typically encompass techniques such as cultivating gratitude, promoting acts of kindness, compassion, character strengths, mindfulness and acceptance and commitment therapy[22]. Several pairwise meta-analyses have sought to assess the pooled effectiveness of psychological interventions in improving well-being outcomes[19–22]. The findings have shown a range of results, with effect sizes spanning from $r = 0.10$ (ref. 22) to $g = 0.39$ (ref. 20). Notably, one meta-analysis highlighted that mindfulness-based interventions ($g = 0.42$) and multicomponent positive psychological interventions ($g = 0.28$) had the strongest effects[19]. Generally, these psychological interventions tend to show effect sizes within the small to medium range and are influenced by various factors, including the specific target population, the intensity of the intervention and the mode of delivery[19].

However, psychological interventions represent just one approach to promoting well-being, and they are often studied separately from interventions in other domains. One example is evidence indicating that physical activity makes an important contribution to well-being. Meta-analyses have found a medium-sized effect of exercise on subjective well-being ($d = 0.36$)[32], and leisure-time physical activity is also associated with positive affect ($r = 0.21$) and life satisfaction ($r = 0.12$)[25]. Developments in well-being science have also underscored the value of interventions targeting organizations, groups and communities[35], with robust evidence demonstrating the

central role of positive social ties in well-being[27,36,37]. Meta-analyses have reported moderate effect sizes ($g = 0.66$) for social identity-building interventions on well-being[33]. Nature-based interventions are also increasingly recognized as an approach for enhancing well-being, with a growing body of evidence supporting their effectiveness[34,38–41]. Epidemiological evidence indicates that proximity to and time spent in nature can have notable positive effects on mental health outcomes and even mortality rates[42–45]. Meta-analyses have also revealed significant effect sizes for the relationship between nature connectedness and both hedonic ($r = 0.20$) and eudemonic ($r = 0.24$) well-being[34]. Connection to nature is also a trait that is associated with pro-environmental behaviours[46] and nature conservation efforts[47], highlighting opportunities to promote well-being of the individual as well as the planet.

There is now a need to synthesize and compare the efficacy of these widely accepted and prescribed well-being interventions from across disciplines and domains. Prior research has also been constrained by pairwise meta-analyses, which only allows the comparison of two interventions. The aim of the present study is to conduct a systematic review and network meta-analysis (NMA) to investigate the comparative effectiveness of multiple well-being-focused interventions including psychological interventions, exercise, social identity-building and nature-based interventions in a single analysis. We also aim to determine whether interventions which target multiple domains (for example, exercise performed in nature or combined with a psychological intervention) are more effective than those with a single focus. To enhance the scope of this research we focus on general population samples rather than specific clinical groups. We acknowledge the general population may however span individuals who are flourishing, languishing or experiencing subclinical symptoms or psychological distress[48]. This supports our aim of maintaining methodological rigour, in addition to increasing generalizability of findings and identifying scalable, preventative approaches with potential to positively shift the distribution of well-being at scale[49].

## Results

### Results of the search and included studies

Searches returned 9,105 unique studies, of which 183 randomized controlled trials (RCTs) including 22,811 adult participants were used in the final NMA (see Fig. 1 for PRISMA flowchart). The mean age of the participants was 38.30 years (range 18–82 years). Studies took place either at universities (39%), workplaces (27%), communities (22%) or online (12%), and 79% of studies were conducted in western countries. The most frequently reported countries were the USA (25%), China (8%), the UK (7%), Australia (6%) and Spain (5%). Demographic reporting across included trials was incomplete. Only approximately half of studies reported participant sex, typically comprising a slightly higher proportion of women, and very few studies reported participant ethnicity, with inconsistent categories. A table summarizing study characteristics in addition to a list of references can be found in the Supplementary Information.

In terms of individual study bias risk assessment, the randomization process was found to have a low risk of bias in 81% of the studies (criteria 1.0). Only 24% of the studies had a low risk of bias on deviations from intended interventions (criteria 2.0), mainly due to insufficient information regarding trial protocols, and 73% were rated as having a low risk of bias due to missing outcome data (criteria 3.0), whereas only 33% conducted intention-to-treat analysis. A total of 44% were deemed to have a low risk of bias in outcome measurement (criteria 4.0), and only 31% had a low risk of bias in the selection of reported results (criteria 5.0), with many lacking information on a prespecified analysis plan. Overall, 12 studies (7%) were classified as 'low risk', 61 (33%) were categorized as having 'some concerns' and 110 (60%) were classified as 'high risk'. A summary table of risk of bias classifications can be found in the Supplementary Information.

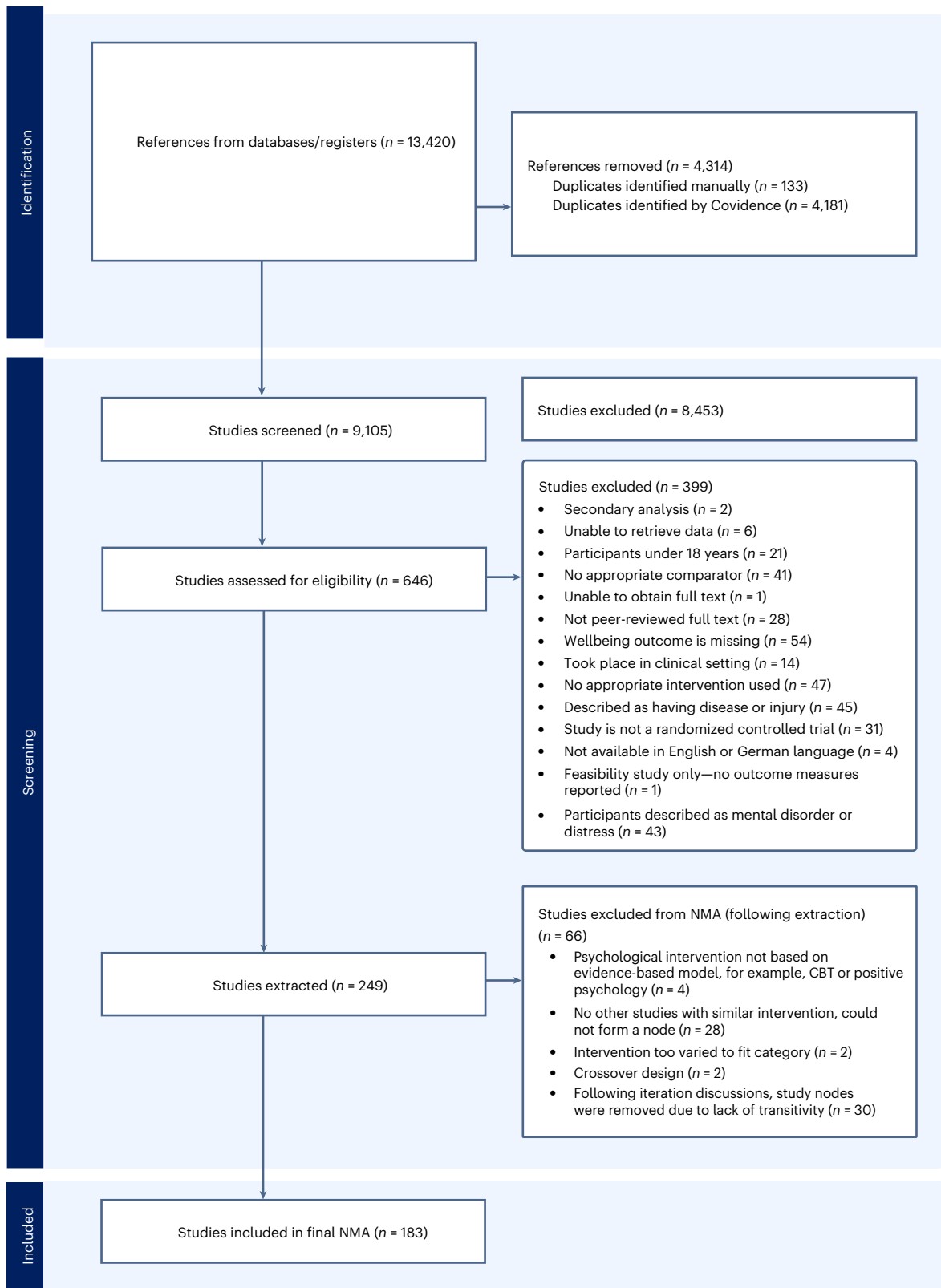

**Fig. 1 | PRISMA flow diagram of the study selection.**

## Network geometry

The most frequently reported active interventions were mindfulness-based Interventions ($n = 72$), followed by exercise-based interventions ($n = 33$) and combined theoretical psychological interventions ($n = 25$). Table 1 describes the node labels and the number of study arms included for each intervention.

Some studies ($n = 66$) or study arms ($n = 34$) were excluded from the NMA following data extraction. For example, on occasion, interventions being compared across multiple arms of the same RCT were not distinct enough to be classified as separate nodes (for example, full versus partial interventions or three good things versus gratitude intervention). In these instances, intervention arms which most

**Table 1 | Summary of intervention nodes included in the NMA**

| Node label | Intervention brief description | N study arms included |
|---|---|---|
| C | No intervention control—includes passive control (for example, sit still), no intervention and waitlist and treatment as usual | 162 |
| MIND | Mindfulness-based approaches | 72 |
| EX | Exercise-based intervention | 33 |
| COMB | Multitheoretical psychological intervention (for example, combination of CBT, PPI and mindfulness). Clear psychological paradigms combined into one | 25 |
| PPI | Single PPI (for example, three good things, character strengths or best possible self) | 20 |
| COMPAS | Compassion-focused therapy | 17 |
| ED | Educational programme or resources (for example, psychoeducation or health behaviour education) | 15 |
| YOGA | Yoga | 14 |
| MPPI | Multicomponent PPI | 8 |
| ACT | Acceptance and commitment therapy | 5 |
| NAT | Nature interventions | 4 |
| EXPSY | Physical movement combined with a psychological intervention (excludes yoga) | 3 |

closely fitted the other interventions in an existing node were included, whereas others were excluded. For example, for the single PPIs node, 'three good things' or 'character strength' intervention arms were chosen over less-standard PPIs, such as 'three funny things'. When an intervention did not clearly fit into any node categories and there were not enough studies to create a new, distinct node, the arm or full study was excluded from NMA, for example, 'mindful-compassion art-based therapy'.

Following assessment of transitivity and local inconsistency, further nodes (for example social interventions, reminiscence interventions and cognitive behavioural therapy (CBT)) had to be excluded from NMA. In addition, some nodes had to be redefined using a tighter description to reduce heterogeneity (see Supplementary Information for a detailed rationale of network geometry adaptations made based on initial transitivity assessments).

The final network (Fig. 2) contained 183 studies, 28 direct comparisons, 38 indirect comparisons and 12 interventions. The network was well connected and had only one subnetwork. Acceptance and commitment therapy (ACT) and nature-based interventions (NAT) were not very strongly attached to the network, as they were compared with control conditions only.

## Transitivity and consistency

For transitivity, a visual inspection of the distribution of potential effect modifiers (Supplementary Information) indicated that, occasionally, characteristics (for example setting, intensity, delivery and format) were sometimes distributed differently across comparisons in the network. However, overall, the variations in the distribution of effects were not substantial, prompting us to proceed with a statistical examination of inconsistency.

In the final assessment of local inconsistency using the node splitting method (SIDE) (Supplementary Information), no comparisons were statistically significant, indicating no inconsistency between direct and indirect estimates.

For the assessment of global inconsistency, the analysis found that there was inconsistency in the network ($\tau^2 = 0.114$; $\tau = 0.338$; $I^2 = 74.2\%$,

$Q = 65.59$, $P < 0.001$), meaning the results varied across different comparisons. However, when a random-effects model was used (allowing full design-by-treatment interactions), the inconsistency dropped and was no longer statistically significant ($Q = 16.53$, degrees of freedom of 27, $P = 0.942$). This suggests that the random-effects model helped account for inconsistency between studies in the network.

## NMA results

In the NMA, nearly all active interventions (except nature-based interventions) significantly outperformed control, with no consistent evidence that psychological approaches (for example, mindfulness, compassion and PPIs) differed significantly from one another (Fig. 3). The only significant comparison between active interventions showed physical exercise combined with psychological intervention (EXPSY) being superior to nature-based interventions (standardized mean difference (s.m.d.) of 0.69, 95% confidence interval (CI) 0.06 to 1.31) (Table 2).

## Treatment ranking

According to $P$-score estimates (Fig. 4), physical exercise combined with psychological intervention (EXPSY), yoga (YOGA), mindfulness (MIND), compassion-based interventions (COMPAS), exercise (EX), single PPIs (PPI), multitheoretical psychological interventions (COMB) and acceptance and commitment therapy (ACT) were ranked as the most effective treatments.

## Additional analyses using meta-regression

We examined whether study or intervention characteristics moderated well-being outcomes using univariate mixed-effect meta-regression analysis. Intervention intensity significantly moderated effects ($F(3, 167) = 4.38$, $P = 0.005$). Medium-length interventions (5–8 weeks) produced significantly stronger effects than short interventions (2–4 weeks) ($\beta = 0.35$, 95% CI 0.56 to 0.15, $P < 0.001$). No evidence of moderation was found for delivery mode ($F(3, 163) = 0.47$, $P = 0.70$), format ($F(1, 167) = 0.65$, $P = 0.42$), study setting ($F(3, 166) = 1.14$, $P = 0.33$), country ($F(1, 168) = 0.98$, $P = 0.32$) or mean participant age ($\beta = 0.002$, 95% CI −0.004 to 0.007, $P = 0.62$). Full model coefficients and 95% confidence intervals are provided in Supplementary Table 3.1.

To examine whether moderator effects differed across intervention types, we conducted exploratory intervention × moderator interaction models that were restricted to four well-represented interventions (mindfulness, exercise, combined psychological interventions and single PPIs).

No significant interactions were detected for delivery mode ($F(7, 106) = 0.15$, $P = 0.99$), format ($F(7, 116) = 0.15$, $P = 0.99$) or study country ($F(7, 117) = 0.16$, $P = 0.99$). For intervention intensity, the overall interaction model reached significance ($F(12, 110) = 2.27$, $P = 0.013$; $R^2 = 11.9\%$). However, most contrasts were non-significant with wide confidence intervals. The only statistically significant effect suggested that medium-length exercise interventions were less effective than short exercise interventions ($\beta = -1.18$, 95% CI −2.25 to −0.12, $p = .03$). Given the sparse data, this isolated finding should be interpreted cautiously and regarded as exploratory. Full model results, including all interaction estimates and 95% CIs, are provided in Supplementary Table 3.2.1.

## Sensitivity analysis using alternative NMA models

Full sensitivity analyses are provided in Supplementary Information 4. Moderator-based sensitivity checks using mixed-effects meta-regressions showed no significant moderation by outcome measure type ($F(3, 169) = 1.08$, $P = 0.36$), control group type ($F(1, 170) = 0.00$, $P = 0.98$) or risk-of-bias category ($F(2, 170) = 1.78$, $P = 0.17$) (Supplementary Table 4.1.1).

Full NMA analyses were also conducted (see the Supplementary Information for full sensitivity analysis results) using four alternative models: (1) excluding studies with high risk of bias, (2) using

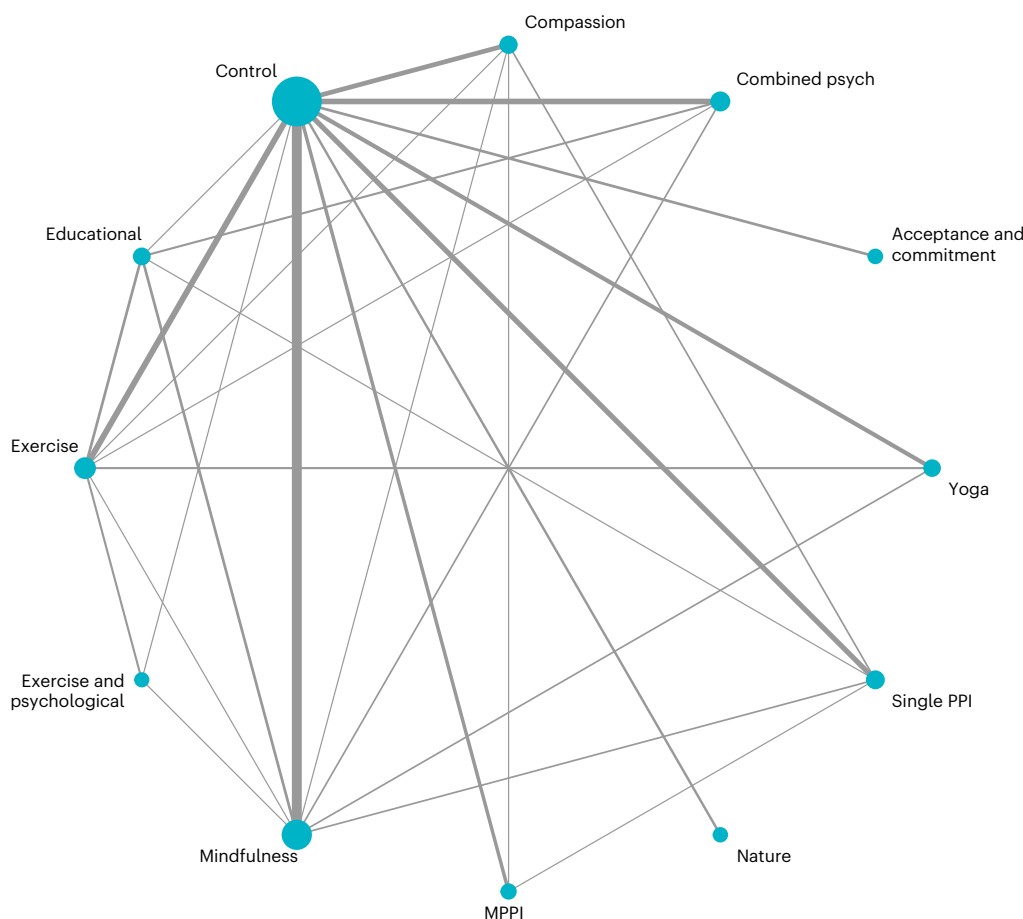

**Fig. 2 | Network plot of intervention comparisons.** The node size reflects the number of participants, and the edge thickness indicates the number of direct trials comparing the two connected interventions. 'C' refers to control arms.

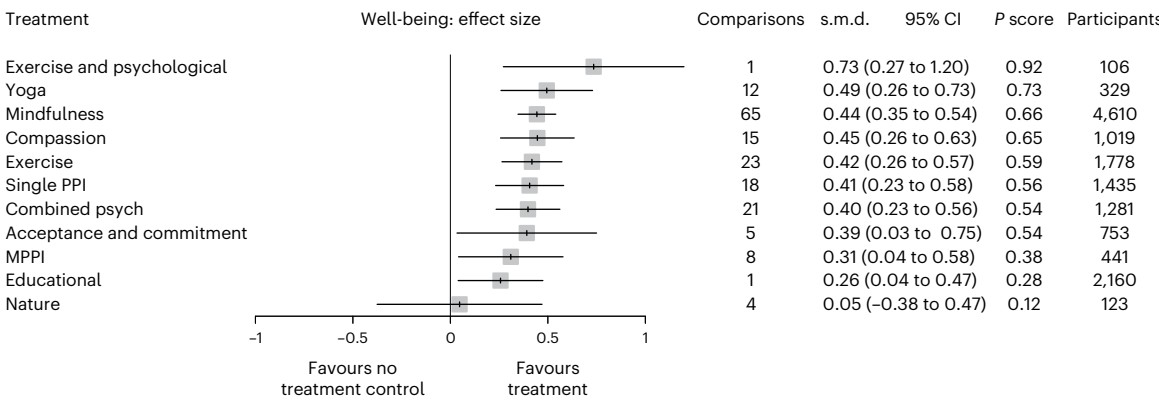

**Fig. 3 | Forest plot of NMA pooled estimated effect sizes for each intervention (NMA primary model).** The s.m.d. with 95% confidence intervals are shown. The interventions are ranked by *P* score (higher is more effective).

subjective well-being as only outcome measure, (3) excluding small studies (defined as studies with included arms totalling an *N* size smaller than the lower quartile of included studies: *n* = 45) and (4) with the inclusion of grey literature. Exercise with psychological intervention (EXPSY), yoga (YOGA) mindfulness (MIND) and compassion (COMPAS) were consistently ranked in the top five interventions across all sensitivity analyses.

When analyses were restricted to studies using subjective well-being as the sole outcome, results were highly consistent with the main NMA: most treatment rankings remained stable, and the same

interventions remained significantly more effective than controls. Excluding small-sample studies (*n* < 45 per trial) yielded findings consistent with the main analysis, with all primary intervention rankings preserved and effect sizes varying minimally. In the model excluding studies at high risk of bias, multicomponent PPIs (MPPI) were no longer statistically significant compared with controls (s.m.d. = 0.26, 95% CI −0.12 to 0.63, *P* = 0.19).

ACT was the only intervention showing substantial variation in ranking across sensitivity analyses. Its estimated effect size increased from s.m.d. = 0.39 (95% CI 0.03 to 0.75) in the main

**Table 2 | League table comparing effect sizes from network pooled estimate versus direct evidence**

| Treatment | ACT | C | COMB | COMPAS | ED | EX | EXPSY | MIND | MPPI | NAT | PPI | YOGA |
|---|---|---|---|---|---|---|---|---|---|---|---|---|
| ACT | ACT | **0.39 (0.03 to 0.75)** | – | – | – | – | – | – | – | – | – | – |
| C | **0.39 (0.03 to 0.75)** | C | **−0.40 (−0.58 to −0.22)** | **−0.47 (−0.67 to −0.26)** | −0.11 (−0.87 to 0.65) | **−0.38 (−0.56 to −0.20)** | −0.83 (−1.72 to 0.07) | **−0.45 (−0.55 to −0.34)** | **−0.31 (−0.59 to −0.03)** | −0.05 (−0.47 to 0.38) | **−0.40 (−0.59 to −0.22)** | **−0.44 (−0.70 to −0.18)** |
| COMB | −0.01 (−0.40 to 0.39) | **−0.40 (−0.56 to −0.23)** | COMB | – | 0.11 (−0.32 to 0.55) | −0.15 (−0.91 to 0.61) | – | 0.03 (−0.58 to 0.63) | – | – | – | – |
| COMPAS | −0.05 (−0.46 to 0.35) | **−0.45 (−0.63 to −0.26)** | −0.05 (−0.30 to 0.20) | COMPAS | – | 0.07 (−0.67 to 0.81) | – | 0.07 (−0.73 to 0.87) | −0.08 (−0.89 to 0.73) | – | −0.17 (−0.71 to 0.37) | – |
| ED | 0.13 (−0.28 to 0.55) | **−0.26 (−0.47 to −0.04)** | 0.14 (−0.10 to 0.39) | 0.19 (−0.10 to 0.47) | ED | −0.27 (−0.63 to 0.09) | – | −0.09 (−0.44 to 0.26) | – | – | −0.01 (−0.92 to 0.91) | – |
| EX | −0.03 (−0.42 to 0.36) | **−0.42 (−0.57 to −0.26)** | −0.02 (−0.23 to 0.19) | 0.03 (−0.21 to 0.26) | −0.16 (−0.39 to 0.07) | EX | −0.32 (−0.80 to 0.16) | −0.13 (−1.02 to 0.76) | – | – | – | −0.17 (−0.68 to 0.34) |
| EXPSY | −0.34 (−0.93 to 0.24) | **−0.73 (−1.20 to −0.27)** | −0.34 (−0.82 to 0.15) | −0.29 (−0.79 to 0.21) | −0.48 (−0.98 to 0.02) | −0.32 (−0.77 to 0.13) | EXPSY | 0.17 (−0.72 to 1.05) | – | – | – | – |
| MIND | −0.05 (−0.42 to 0.32) | **−0.44 (−0.54 to −0.35)** | −0.05 (−0.23 to 0.14) | 0.00 (−0.21 to 0.21) | −0.19 (−0.41 to 0.03) | −0.03 (−0.20 to 0.15) | 0.29 (−0.18 to 0.76) | MIND | – | – | 0.16 (−0.47 to 0.79) | 0.04 (−0.65 to 0.73) |
| MPPI | 0.08 (−0.36 to 0.53) | **−0.31 (−0.58 to −0.04)** | 0.09 (−0.23 to 0.40) | 0.14 (−0.18 to 0.46) | −0.05 (−0.40 to 0.29) | 0.11 (−0.20 to 0.42) | 0.43 (−0.11 to 0.96) | 0.13 (−0.15 to 0.42) | MPPI | – | −0.18 (−0.90 to 0.55) | – |
| NAT | 0.34 (−0.21 to 0.90) | −0.05 (−0.47 to 0.38) | 0.35 (−0.10 to 0.80) | 0.40 (−0.06 to 0.86) | 0.21 (−0.27 to 0.69) | 0.37 (−0.08 to 0.82) | **0.69 (0.06 to 1.31)** | 0.40 (−0.04 to 0.83) | 0.26 (−0.24 to 0.76) | NAT | – | – |
| PPI | −0.01 (−0.41 to 0.38) | **−0.41 (−0.58 to −0.23)** | −0.01 (−0.25 to 0.23) | 0.04 (−0.21 to 0.29) | −0.15 (−0.42 to 0.12) | 0.01 (−0.22 to 0.24) | 0.33 (−0.17 to 0.82) | 0.04 (−0.16 to 0.21) | −0.10 (−0.41 to 0.21) | −0.36 (−0.82 to 0.10) | PPI | – |
| YOGA | −0.10 (−0.53 to 0.33) | **−0.49 (−0.73 to −0.26)** | −0.10 (−0.38 to 0.19) | −0.05 (−0.35 to 0.25) | −0.24 (−0.55 to 0.08) | −0.08 (−0.34 to 0.19) | 0.24 (−0.27 to 0.75) | −0.05 (−0.30 to 0.20) | −0.18 (−0.54 to 0.17) | −0.45 (−0.93 to 0.04) | −0.09 (−0.38 to 0.20) | YOGA |

The upper triangle represents the direct pairwise meta-analyses (s.m.d. and 95% CIs). The lower triangle represents the NMA estimates (pooled estimates from direct and indirect evidence) (s.m.d. and 95% CIs). The bolded values are the statistically significant differences ($P < 0.05$).

model to an s.m.d. of 0.50 (95% CI 0.19 to 0.81; second rank) in the subjective-well-being-only model and s.m.d. of 0.51 (95% CI 0.24 to 0.78; fifth rank) in the low–medium-risk-of-bias model. However, when small studies were excluded, ACT was no longer statistically different from control (s.m.d. of 0.22, 95% CI −0.07 to 0.50; tenth rank). These shifts suggest that ACT estimates were most affected by study quality and sample size. Possible causes of these discrepancies are explored in the discussion.

In addition, we conducted a sensitivity analysis including seven unpublished trials identified through targeted grey literature searches (one unpublished clinical trial and six dissertations/theses), adding a further 709 participants across mindfulness, compassion, exercise and ACT interventions (Supplementary Information). These studies met all other eligibility criteria. The results were highly consistent with the primary NMA: effect size estimates shifted only minimally (≤0.03), no interventions changed statistical significance and only minor shifts were observed in ranking.

### Certainty of evidence

Regarding certainty of evidence, 71% of comparisons were rated as moderate, 26% of comparisons were rated as low and 3% comparisons were rated as very low. The certainty of evidence for each network estimate is reported in the Supplementary Information. There is no clear trend regarding which intervention comparisons contain low or very low certainty of evidence, suggesting bias is moderately evenly distributed.

### Publication bias

The funnel plot of direct comparisons versus control showed asymmetry, with effect sizes tending to increase as standard error rose (Supplementary Information). Egger's regression test confirmed significant funnel plot asymmetry both when only published studies were considered ($t(206) = −4.16$, $p < 0.001$, bias estimate of −1.20, s.e. of 0.29) and when grey literature was added ($t(213) = −3.98$, $P < 0.001$, bias estimate of −1.14, s.e. of 0.29). The inclusion of grey literature slightly attenuated the estimated bias but did not alter the overall conclusion. However, given that sensitivity analyses excluding studies with high risk of bias and small sample sizes yielded results highly consistent with the main network model, overall conclusions were unlikely to have been substantially influenced by publication bias.

## Discussion

This NMA aims to advance the understanding of the relative impacts of well-being interventions, a topic of great research interest and debate[20,50–52]. Our analysis synthesized evidence from 183 RCTs on psychological interventions, exercise and nature-based interventions into a single analytical framework. Interventions combining exercise and psychological interventions (EXPSY) showed the highest effect size (s.m.d. of 0.73, 95% CI 0.27 to 1.20), although based on few studies with high confidence intervals. Yoga, mindfulness and compassion-based and positive psychological interventions (s.m.d. of 0.41–0.49) all demonstrated consistent high rankings and moderate effect sizes with

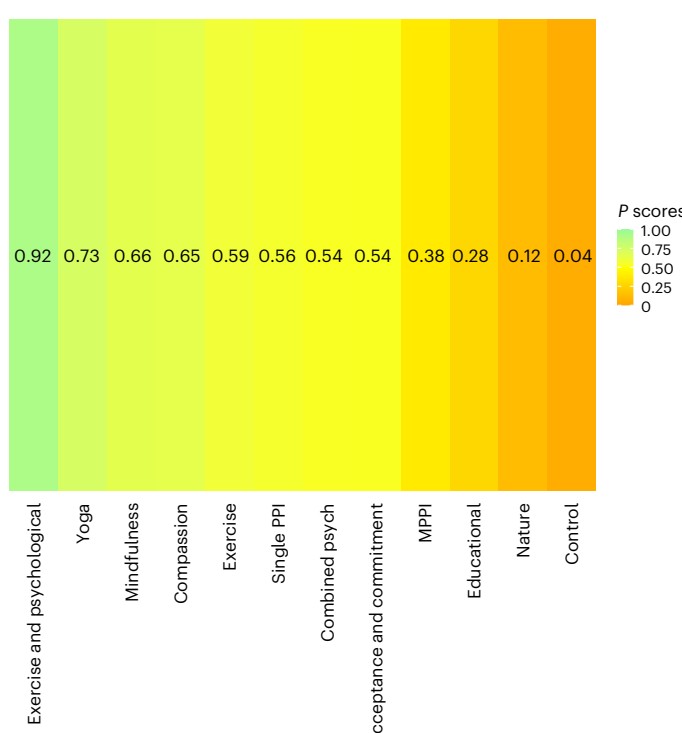

**Fig. 4 | Intervention ranking based on P scores.** The bar lengths and colours indicate relative intervention ranking from the NMA (a higher P score indicates a higher ranking).

greater precision. Exercise also showed comparable effects (s.m.d. of 0.42, 95% CI 0.26 to 0.57) to conventional psychological interventions.

The combination of exercise and psychological interventions node (EXPSY) included three interventions: awe walks[53], meditation combined with brisk walking[54] and walking groups with positive psychology coaching[55]. This finding should be interpreted with caution as out of only three studies, two have a high risk of bias, and the pooled effect size showed wide confidence intervals (s.m.d. of 0.73, 95% CI 0.27 to 1.20). Only one of these studies provided a direct comparison with control, but as this was a brief 10-min intervention in young adults, its generalizability is limited[54]. Most comparisons, however, were rated as moderate in certainty using CINEMA. All interventions involved walking, which minimized heterogeneity between interventions but restricts generalizability to other exercise types. Despite these limitations, the consistent high ranking of the node across analyses suggests that integrated psychological interventions with physical exercise hold promise and should be prioritized in future RCTs using longer-term interventions, varied exercise modalities and different psychological techniques.

Exercise alone also showed a moderate effect size (s.m.d. of 0.42, 95% CI 0.26 to 0.57) comparable to those of established psychological approaches (for example, positive psychological and compassion-based interventions). This should not be interpreted as evidence that movement can substitute for psychological care, rather, it suggests multiple viable pathways exist for enhancing well-being, supporting personalized intervention approaches that consider individual preferences, capabilities and circumstances. This robust effect also further supports the rationale for combining exercise with psychological approaches.

Mind–body interventions demonstrated consistently strong effects. Mindfulness (s.m.d. of 0.44; 95% CI 0.35 to 0.54) was supported by the most extensive evidence base: 65 direct head-to-head comparisons and relatively stable effects across sensitivity analyses. Yoga also ranked highly, though the evidence base was less precise (s.m.d.

of 0.49; 95% CI 0.26 to 0.73) and derived from fewer participants and comparisons. Yoga is traditionally understood as encompassing more than physical postures (āsana), typically integrating dhyāna (meditative practice) and prāṇāyāma (breath regulation)[56], which situates it conceptually close to mindfulness interventions. Mindfulness is also a core component of compassion-focused interventions, and similarly, compassion is explicitly targeted in most structured mindfulness courses, as both approaches stem from Buddhist philosophy where they are seen as mutually dependant practices[57,58]. Across diverse modalities, the cultivation of mindfulness (whether through meditation, present-moment awareness, fostering acceptance or embodied practices) emerged as a shared feature. Rather than being viewed as competing, these findings suggest this family of contemplative interventions collectively hold promise for promoting well-being and reinforce the large body of work supporting the effectiveness of mindfulness-based approaches[59–61].

The variability in ACT's effectiveness across sensitivity analyses was largely driven by one study: Danitz[62]. This high-risk-of-bias trial measured well-being with the Philadelphia Mindfulness Scale, which includes 'awareness' and 'acceptance' subscales. We extracted the 'awareness' subscale, which aligned more closely with our 'positively framed' definition of well-being, but ACT only significantly improved 'acceptance', while baseline imbalances biased the awareness scores in favour of the control group, artificially lowering ACT's effect. Danitz[62] was excluded from sensitivity analyses removing high-risk which raised ACT's effect size from s.m.d. of 0.37 to 0.46 and improved its ranking from seventh to fourth. However, in further analysis that excluded small studies, the Danitz study remained in the analysis, suppressing the overall estimate and rendering it non-significant. Importantly however, our main effect estimate (s.m.d. of 0.39; 95% CI 0.03 to 0.75) aligns similarly with that from a large (N = 1,162) ACT trial[63] with a low risk of bias (d = 0.37, 95% CI 0.26 to 0.49). Therefore, we consider our main NMA estimate for ACT and its relative effectiveness to be valid. One interpretation could be that ACT's comparatively smaller effect on well-being may potentially reflect its stronger cognitive focus and less emphasis on sustained contemplative, embodied or positive affect-generating practices that characterize mindfulness, yoga and compassion-based interventions.

MPPIs had a smaller effect size and ranked lower in our NMA than individual PPIs, with the 'Three Good Things' gratitude intervention being prominently featured within the single PPI category. This contradicts the findings of a large meta-analysis[19], which favoured MPPIs over single PPIs. A potential explanation for this is that our analysis included a study which directly compared a PPI with a MPPI and reported no significant difference in their efficacy on well-being[64]. The slightly larger effect size estimate for single PPIs in our analysis, however, should not necessarily support a clinical preference for them over MPPIs. We found no statistically significant difference between single and MPPIs (Table 2) and overlapping confidence intervals suggest contextual factors will probably play a vital role in intervention selection[65].

Contrary to expectations, nature-based interventions did not significantly outperform controls. However, this finding should be interpreted cautiously. The nature-based node, similarly to ACT, was weakly integrated within the broader network and its estimate was based largely on indirect evidence, as trials only compared against control conditions. Studies in this group were small, at moderate-to-high risk of bias and conceptually diverse, ranging from horticultural therapy to nature photography. While all took place in natural environments, they varied widely in their psychological aims, delivery formats and therapeutic mechanisms. This conceptual heterogeneity probably diluted the pooled effect estimate and limits interpretability. To improve clarity in future research, we suggest defining nature-based interventions not merely by setting but by whether they actively cultivate nature connectedness as a psychological mechanism. Growing evidence suggests that interventions designed to deepen emotional and sensory engagement

with nature, rather than simply providing exposure are more effective in promoting well-being and encouraging pro-environmental behaviour[46,66]. Future trials and reviews should explicitly distinguish between these approaches.

Our protocol defined social identity-building interventions as those aiming to form or strengthen a shared sense of group identity or belonging, grounded in social identity theory[36]. In practice, however, identifying suitable RCTs proved challenging. Few studies explicitly referenced 'social identity', and those that were potentially eligible varied widely in content including discussions of current events[67], reminiscence groups[68], parenting workshops[69] and emotional peer support[70], making it difficult to define a conceptually coherent node. Moreover, nearly all studies in this category were conducted with older adult populations, whereas other intervention types typically involved mixed-age samples. Given that NMA assumes a comparable distribution of effect modifiers across nodes, this demographic imbalance violates the assumption of transitivity. For these reasons, we excluded social identity-building interventions from the final network. Future research would benefit from more explicit operationalization of social identity mechanisms and clearer reporting of group dynamics within interventions.

Regarding potential limitations, the strength of our NMA depends on the quality of the included RCTs and is therefore shaped by limitations common to well-being intervention research. Many studies provided only immediate postintervention outcomes, thus long-term follow-up effects could not be analysed due to sparse data. Moreover, we observed a high risk of bias in many trials. Only 33% of studies employed intention-to-treat analyses, and we observed asymmetry in the funnel plot, suggesting potential publication bias. To mitigate these concerns and enhance reliability, we restricted inclusion to RCT designs and validated well-being measures. Sensitivity analyses (Supplementary Information), including the exclusion of high-risk and small sample studies, the addition of grey literature trials and models restricted to subjective well-being outcomes, all produced results consistent with the main model. This consistency across all sensitivity checks increases confidence in the findings.

In addition, since few interventions across domains had been tested head-to-head, many comparisons in our network relied on indirect evidence. While this is a strength of NMA (allowing comparisons across interventions without direct trials), the overall strength of evidence remains dependent on the quality and connectedness of studies, which varied across intervention nodes. Critically however, the assumption of transitivity was met and no significant inconsistency between direct and indirect estimates in the final network model was found (Supplementary Information), strengthening confidence in our pooled estimates.

We initially considered a component NMA[71,72] to examine the additive effects of intervention components (for example, mindfulness and exercise). However, this approach was not viable due to limited component overlap, poor reporting of intervention content and a lack of multicomponent designs. For example, only three studies met criteria for the 'exercise and psychological' intervention node, each with differing content. These limitations led us to adopt a standard NMA with designated nodes for multicomponent interventions where feasible. Future studies which explore multicomponent and cross-discipline interventions with clearer reporting could support the component NMA and help identify active ingredients across intervention types.

By targeting general population samples rather than clinical groups, we focus on scalable well-being interventions that can benefit individuals across the entire well-being spectrum. The methodological decision to exclude clinical samples, while preserving network transitivity assumptions, means our findings are applicable to universal well-being promotion efforts from supporting those experiencing subclinical distress to optimizing well-being in already flourishing individuals. This broad applicability, combined with our transdisciplinary approach spanning psychological, physical and environmental interventions, offers evidence for preventive mental health strategies

**Table 3 | Eligibility criteria based on PICOS framework**

| | Inclusion criteria | Exclusion criteria |
|---|---|---|
| Population | Adults >18 years old | Participants under 18 years old Participants described as having a specific condition, disease or dysfunction |
| Intervention | Psychological interventions, exercise, social support or nature-based interventions. Can be online, in-person or hybrid | Pharmacological or drug treatment arms |
| Comparison | A randomized controlled 'eligible' intervention or no intervention or a waitlist control | No randomly assigned control condition |
| Outcome | Well-being (primary outcome) | Single item measures of well-being. Studies which solely define well-being as reduction of ill-being (for example reduced anxiety scores) |
| Study type | RCTs | Observational studies, non-randomized trials and unpublished trials |

at a time when global challenges increasingly demand resilient and adaptive populations capable of thriving amid uncertainty.

## Conclusion

This NMA offers a synthesis of well-being-focused interventions from 183 RCTs spanning psychological, physical, environmental and integrative approaches. Our findings support the effectiveness of established psychological interventions including mindfulness, compassion-based therapies and positive psychology approaches. The comparable effectiveness of exercise and yoga with traditional psychological well-being interventions highlights multiple potential pathways to well-being promotion, underscoring the importance of interdisciplinary thinking in both research and policy. Interventions combining physical activity with psychological strategies showed the largest effects, though this finding is based on only three studies and requires replication in adequately powered trials. The consistent effectiveness of mind–body interventions such as mindfulness, compassion-focused interventions and yoga suggests these interventions share components such as contemplative practice and embodied regulation that support well-being. Our main conclusions remained stable across multiple sensitivity analyses. Persistent methodological and conceptual limitations in well-being science highlight the need for future RCTs to be well-powered, transparently reported and for interventions to integrate transdisciplinary dimensions of well-being. The next step is to continue to translate intervention evidence from well-being and contemplative sciences into policies and systems that promote equitable, population-level access to effective approaches for enhancing well-being.

## Methods
### Protocol and registration
This review was preregistered in March 2023 on PROSPERO (CRD42023403480), where the study design, inclusion criteria, outcome measures and use of NMA were prespecified. No deviations from the registered protocol occurred. The development and refinement of intervention nodes based on the data extracted are fully documented in the Supplementary Information.

### Selection of studies and data extraction
We conducted systematic searches of MEDLINE, PsycINFO, CENTRAL and Scopus from database inception to March 2023. Search terms

combined keywords and subject headings related to well-being (for example, well-being, positive affect and resilience), interventions (for example, mindfulness, exercise, yoga, nature and positive psychology) and study design (for example, randomized controlled trial and clinical trial). Boolean operators (AND, OR) and database-specific filters for human participants and adults were applied. Reference lists of eligible studies and relevant reviews were also screened for additional trials. Full search strategies for each database are provided in the Supplementary Information.

References were managed using Covidence (https://www.covidence.org/). Titles and abstracts were screened independently by three reviewers (A.G., I.G. and S.K.). Those titles and abstracts which resulted in disagreements were automatically included for full-text screening. Reviewers also screened the full texts according to eligibility criteria and recorded reasons for exclusions. Disagreements during full text screening were resolved by discussion or by the first author (L.W.). The data were extracted using a custom form on Covidence which was then checked for accuracy by L.W.

Data extraction included detailed information about study and participant characteristics, intervention content and outcomes. For each trial, data were recorded for up to four intervention arms, capturing delivery mode (for example, in-person, online and self-guided), format (group, individual or both), duration, frequency and total number of sessions, along with a brief description of content. Participant data included sample size, mean age, gender distribution, recruitment method and any exclusion criteria related to mental or physical health.

Pre- (baseline) and postintervention well-being scores were used; follow-up data were also extracted where reported. Where studies assessed well-being through multiple measures (for example, subjective well-being, positive affect and resilience), all eligible outcomes were extracted, and subjective well-being was prioritized for synthesis. The preferred choice of extraction for outcome measures were means and standard deviations. When possible, these were calculated using alternative reported statistics, or study authors were contacted (and followed up at least once) via email to request missing data. Studies were excluded if no response was received by time of data analysis. Methodological quality of included randomized control studies were assessed using the revised Cochrane risk-of-bias tool for randomized trials (RoB 2) by at least one reviewer and was checked for accuracy by the first author. Any disagreements were resolved via discussion with wider review team.

### Eligibility criteria

Eligibility criteria were developed using the PICOS framework and are summarized in Table 3. We included RCTs (both parallel and cluster) published in peer-reviewed academic journals. A supplementary sensitivity analysis was later conducted including additional grey literature trials (Supplementary Material). Publication date could be any time before search. Participants had to be aged 18 years or older and not described as having a diagnosable condition, disease or dysfunction or receiving medical treatment for a disease at the time of study. Studies had to deliver at least one intervention described as being a psychological intervention, exercise intervention, social identity or social support intervention, a nature-based intervention or contain a combination of these. They could be either individual or group format and could be delivered face to face, online or hybrid. Every study arm was assessed independently against PICOS eligibility criteria (Table 3). Interventions could either be compared with a second eligible intervention or a no intervention or waitlist control group. Studies in English or German language could be included.

### Outcome measures

The primary outcome for this study was psychological well-being, defined as the presence of positive or adaptive characteristics such as measures of subjective well-being, life satisfaction, happiness, positive affect, resilience or flourishing. Acknowledging that these characteristics may vary, 'category of well-being outcome measure used' was included as a sensitivity analysis to determine that measures did not lead to different findings. Studies which solely measured 'well-being' as a reduction in ill-being (for example, reduced depression/anxiety scores) were excluded. Common standardized well-being scales included Perceived Wellness Score, Warwick–Edinburgh Mental Wellbeing Scale (WEMWBS), 36-Item Short-Form Health Survey (SF-36): Mental Component Subscale, World Health Organization Wellbeing Scale (WHO-5) and PANAS-Positive Affect. This list of scales is non-exhaustive, and other measures were included if they met inclusion criteria.

### Network geometry and nodes

The network nodes were defined following discussion with research team members including clinicians with expertise on which interventions could logically be clustered together, based on both their underpinning theory and delivery in practice.

### Software

All analyses were conducted in R (version 4.4.2) using the packages netmeta, meta, metafor, dplyr and ggplot2. Network meta-analyses were implemented using the netmeta package, and confidence in estimates was assessed using CINeMA (https://cinema.ispm.unibe.ch/). All code are available in the project's Open Science Framework (OSF) repository (https://osf.io/nz59j/?view_only=30f14278418f454e8c6ee297493f2c39).

### Statistical analysis

The assumption of transitivity requires all interventions to be jointly randomizable. If this assumption holds, common comparisons should not vary substantially on key characteristics or the validity of the indirect comparisons will be questionable (for example the 'A' in 'A v B' should not differ from the 'A' in 'A v C', otherwise the indirect 'B v C' comparison will be invalid). To assess transitivity, we created a table of important characteristics (study setting, intervention intensity and delivery mode) to examine whether potential effect modifiers were similarly distributed across the comparisons. Variability from different study populations, interventions and outcomes makes it difficult to ascertain that the treatments are being compared under equivalent conditions. The inclusion of heterogeneous interventions necessitates even more stringent control of population differences. Hence, we opted to use a general population sample, to ensure more uniformity and reliable comparisons.

For pairwise meta-analyses, we conducted pairwise meta-analyses for all direct comparisons using a random-effects model. Homogeneity of effect sizes were estimated using $\tau^2$ and Higgins $I^2$ values. The s.m.d.s were reported with 95% confidence intervals. The $P$ values (alpha threshold of 0.05) were used to determine whether the effect sizes for each direct comparison were significant.

For NMA, a random-effect NMA was conducted to estimate a single summary effect for each node in the network. The NMA synthesized both direct (head-to-head) and indirect comparisons across trials to generate pooled effect size estimates of s.m.d.s between interventions. This approach enables the comparison of multiple interventions simultaneously, even when few direct comparisons exist and strengthens the precision of effect estimates by incorporating all available evidence.

Global inconsistency was assessed using the $Q$ statistic, based on design-by-treatment interaction model[73]. Local inconsistency was assessed by comparing direct estimates to indirect estimates using the node splitting method (SIDE) (whereby $P < 0.1$ indicated statistically significant inconsistency). Treatments were ranked using $P$ scores, these range from 0 to 1 and can be interpreted as an average degree of certainty for a treatment to be better than the other treatments in the network[74].

Small study effects were assessed using comparison-adjusted funnel plots, which report each study's effect estimate against their

reversed standard error. Asymmetry in the plot suggests that larger effects tend to be systematically found in smaller studies.

In terms of additional analyses, exploratory moderator analyses were conducted to examine whether the pooled effect of all active interventions versus control varied across prespecified effect modifiers. Analyses were implemented as mixed-effects meta-regressions, which do not rely on assumptions of network connectivity or transitivity.

The following moderators were tested:

- Delivery mode (in-person, online platform, live video conferencing or self-guided)
- Treatment format (individual versus group)
- Intervention intensity (Brief, short, medium or long—based on weeks)
- Study setting (university, workplace, community or online/other)
- Country (western versus non-western)
- Age
- Type of well-being outcome measure (subjective well-being, resilience, mindfulness or positive affect)
- Control condition (waitlist versus no-intervention)
- Risk of bias category (low, medium or high)

In addition, we conducted sensitivity analyses using full NMA models to test our findings under different assumptions. These examined whether the relative effectiveness of interventions differed when: (1) only studies using subjective well-being as an outcome were included, (2) high-risk-of-bias studies were excluded, (3) studies with small sample sizes were excluded and (4) grey literature trials were included. These moderators were selected based on their potential to bias effect estimates and were available across the full network.

The risk of bias across studies was assessed with confidence in NMA (CINeMA) for NMA[75]. CINeMA considers six domains that impact confidence in the NMA results: (1) within-study bias, (2) reporting bias, (3) indirectness, (4) imprecision, (5) heterogeneity and (6) incoherence. Each treatment comparison was assessed as having 'no concerns', 'some concerns' or 'major concerns' in each of the six domains. Then, judgments across the domains were summarized into a single confidence rating (high, moderate, low or very low).

### Reporting summary

Further information on research design is available in the Nature Portfolio Reporting Summary linked to this article.

## Data availability

All data supporting the findings of this study are available via the Open Science Framework (OSF) repository at https://osf.io/nz59j/?view_only=30f14278418f454e8c6ee297493f2c39. The repository includes the dataset extracted from all included trials used for NMA and meta-regression. Source data are provided with this paper.

## Code availability

The code used for this study is available via OSF at https://osf.io/nz59j/?view_only=30f14278418f454e8c6ee297493f2c39. The R script includes all steps for data processing, analysis and visualization.

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

## Acknowledgements

This work received no external funding.

## Author contributions

The study was conceived and designed by LW.,. A.H.K. and Z.F. L.W. led the literature review, data extraction and analysis, supported by A.G., I.G. and S.K. L.W. developed the methodological approach, including analysis scripts, managed project administration and drafted the initial paper. A.G., I.G. and S.K. contributed to data coding and verification under the supervision of L.W. and A.H.K. All authors contributed to paper revision and approved the final version. A.H.K. and Z.F. provided overall supervision and guidance throughout the project.

## Competing interests

L.W. declares a potential competing interest related to their professional work as a yoga and mindfulness teacher. This experience did not influence the design, conduct, analysis or interpretation of the research. The other authors declare no competing interests.

## Additional information

**Correspondence and requests for materials** should be addressed to Lowri Wilkie or Andrew Haddon Kemp.

# Reporting Summary

## Statistics

For all statistical analyses, confirm that the following items are present in the figure legend, table legend, main text, or Methods section.

| n/a | Confirmed | |
|-----|-----------|---|
| ☒ | ☐ | The exact sample size (*n*) for each experimental group/condition, given as a discrete number and unit of measurement |
| ☒ | ☐ | A statement on whether measurements were taken from distinct samples or whether the same sample was measured repeatedly |
| ☐ | ☒ | The statistical test(s) used AND whether they are one- or two-sided<br>*Only common tests should be described solely by name; describe more complex techniques in the Methods section.* |
| ☐ | ☒ | A description of all covariates tested |
| ☒ | ☐ | A description of any assumptions or corrections, such as tests of normality and adjustment for multiple comparisons |
| ☐ | ☒ | A full description of the statistical parameters including central tendency (e.g. means) or other basic estimates (e.g. regression coefficient) AND variation (e.g. standard deviation) or associated estimates of uncertainty (e.g. confidence intervals) |
| ☐ | ☒ | For null hypothesis testing, the test statistic (e.g. *F*, *t*, *r*) with confidence intervals, effect sizes, degrees of freedom and *P* value noted<br>*Give P values as exact values whenever suitable.* |
| ☒ | ☐ | For Bayesian analysis, information on the choice of priors and Markov chain Monte Carlo settings |
| ☐ | ☒ | For hierarchical and complex designs, identification of the appropriate level for tests and full reporting of outcomes |
| ☐ | ☒ | Estimates of effect sizes (e.g. Cohen's *d*, Pearson's *r*), indicating how they were calculated |

*Our web collection on statistics for biologists contains articles on many of the points above.*

## Software and code

Policy information about availability of computer code

| Data collection | Data collection, study screening, and extraction were conducted using Covidence (https://www.covidence.org/). No custom software was developed for data collection. |
|---|---|
| Data analysis | All analyses were conducted in R (version 4.4.2) using the packages netmeta, meta, metafor, dplyr, and ggplot2. Network meta-analyses were implemented using the netmeta package, and confidence in estimates was assessed using CINeMA (https://cinema.ispm.unibe.ch/). All code are available in the project's Open Science Framework (OSF) repository [https://osf.io/nz59j/?view_only=30f14278418f454e8c6ee297493f2c39]. |

For manuscripts utilizing custom algorithms or software that are central to the research but not yet described in published literature, software must be made available to editors and reviewers. We strongly encourage code deposition in a community repository (e.g. GitHub). See the Nature Portfolio guidelines for submitting code & software for further information.

## Data

Policy information about availability of data

All manuscripts must include a data availability statement. This statement should provide the following information, where applicable:

- Accession codes, unique identifiers, or web links for publicly available datasets
- A description of any restrictions on data availability
- For clinical datasets or third party data, please ensure that the statement adheres to our policy

All data supporting the findings of this study are available in the Open Science Framework (OSF) repository at: https://osf.io/nz59j/?view_only=30f14278418f454e8c6ee297493f2c39. The repository includes the dataset extracted from all included trials used for network meta-analysis and meta-regression.

## Research involving human participants, their data, or biological material

Policy information about studies with human participants or human data. See also policy information about sex, gender (identity/presentation), and sexual orientation and race, ethnicity and racism.

| | |
|---|---|
| Reporting on sex and gender | This study is a network meta-analysis that synthesises data from previously published randomised controlled trials (RCTs). No new data were collected from human participants, and therefore, ethical oversight, recruitment, and participant-level demographic reporting do not apply. |
| Reporting on race, ethnicity, or other socially relevant groupings | *Please specify the socially constructed or socially relevant categorization variable(s) used in your manuscript and explain why they were used. Please note that such variables should not be used as proxies for other socially constructed/relevant variables (for example, race or ethnicity should not be used as a proxy for socioeconomic status).* *Provide clear definitions of the relevant terms used, how they were provided (by the participants/respondents, the researchers, or third parties), and the method(s) used to classify people into the different categories (e.g. self-report, census or administrative data, social media data, etc.)* *Please provide details about how you controlled for confounding variables in your analyses.* |
| Population characteristics | See Behavioural & social sciences study design |
| Recruitment | No human participants were recruited |
| Ethics oversight | No ethical approval was required. Only published data was used. |

Note that full information on the approval of the study protocol must also be provided in the manuscript.

# Field-specific reporting

Please select the one below that is the best fit for your research. If you are not sure, read the appropriate sections before making your selection.

☐ Life sciences ☒ Behavioural & social sciences ☐ Ecological, evolutionary & environmental sciences

For a reference copy of the document with all sections, see nature.com/documents/nr-reporting-summary-flat.pdf

# Behavioural & social sciences study design

All studies must disclose on these points even when the disclosure is negative.

| | |
|---|---|
| Study description | This study is a network meta-analysis (NMA) synthesizing quantitative data from previously published randomised controlled trials (RCTs). No new data were collected, and therefore, information regarding sampling, data collection, timing, exclusions, non-participation, and randomisation refers to the original studies included. |
| Research sample | The mean age across studies was 38.3 years (range 18–82). Studies were primarily conducted in Western countries (79%), most frequently the USA, China, the UK, Australia, and Spain. Approximately half of studies reported participant sex, with most indicating a majority of women. Fewer than 10% of studies reported participant ethnicity, and reporting categories were inconsistent. All included trials recruited non-clinical adult samples. |
| Sampling strategy | No new participant sampling was conducted. Sample sizes were determined by the original trials and varied across studies. The final analysis included 183 RCTs comprising 22,811 adult participants. The large aggregated sample provides sufficient statistical power for estimating comparative effects. |
| Data collection | This study did not involve the collection of new participant data. All data were extracted from previously published randomised controlled trials identified through systematic database searches. Screening and data extraction were conducted independently by two reviewers using Covidence, with discrepancies resolved through discussion. Extracted data included study characteristics, intervention details, sample size, and wellbeing outcome measures. As all analyses were based on published summary statistics, no |

| | |
|---|---|
| | participants or researchers were present during data collection, and blinding was not applicable. |
| Timing | The primary literature search was conducted in March 2023 across multiple electronic databases (PsycINFO, PubMed/MEDLINE, Scopus, and Web of Science) using predefined search terms for wellbeing interventions in adult populations. In response to reviewer feedback, an updated search was conducted in August 2025 to identify unpublished or grey literature. This supplementary search included ClinicalTrials.gov, the National Institute for Health and Care Research (NIHR) database, ProQuest Dissertations, and reference lists of relevant reviews. Both searches followed PRISMA 2020 guidelines and used identical inclusion and exclusion criteria. |
| Data exclusions | Of 9,105 unique records screened, 183 randomised controlled trials met the inclusion criteria and were included in the final network meta-analysis (see Table 3 for eligibility criteria and Figure 1 for the PRISMA flow diagram). Following data extraction, 66 studies were excluded during standard network refinement to maintain model assumptions and connectivity (see Supplementary Information 2.2.3). |
| Non-participation | Not applicable. This study is a systematic review and network meta-analysis of previously published trials. No participants were directly recruited or followed by the authors. |
| Randomization | All included studies were randomised controlled trials (RCTs) in which participants were allocated to experimental and control conditions using randomisation procedures as reported in the original studies. No new participant allocation occurred in this network meta-analysis. |

# Reporting for specific materials, systems and methods

We require information from authors about some types of materials, experimental systems and methods used in many studies. Here, indicate whether each material, system or method listed is relevant to your study. If you are not sure if a list item applies to your research, read the appropriate section before selecting a response.

## Materials & experimental systems

| n/a | Involved in the study |
|---|---|
| ☒ | ☐ Antibodies |
| ☒ | ☐ Eukaryotic cell lines |
| ☒ | ☐ Palaeontology and archaeology |
| ☒ | ☐ Animals and other organisms |
| ☒ | ☐ Clinical data |
| ☒ | ☐ Dual use research of concern |
| ☒ | ☐ Plants |

## Methods

| n/a | Involved in the study |
|---|---|
| ☒ | ☐ ChIP-seq |
| ☒ | ☐ Flow cytometry |
| ☒ | ☐ MRI-based neuroimaging |

## Plants

| | |
|---|---|
| Seed stocks | *Report on the source of all seed stocks or other plant material used. If applicable, state the seed stock centre and catalogue number. If plant specimens were collected from the field, describe the collection location, date and sampling procedures.* |
| Novel plant genotypes | *Describe the methods by which all novel plant genotypes were produced. This includes those generated by transgenic approaches, gene editing, chemical/radiation-based mutagenesis and hybridization. For transgenic lines, describe the transformation method, the number of independent lines analyzed and the generation upon which experiments were performed. For gene-edited lines, describe the editor used, the endogenous sequence targeted for editing, the targeting guide RNA sequence (if applicable) and how the editor was applied.* |
| Authentication | *Describe any authentication procedures for each seed stock used or novel genotype generated. Describe any experiments used to assess the effect of a mutation and, where applicable, how potential secondary effects (e.g. second site T-DNA insertions, mosiacism, off-target gene editing) were examined.* |

