## [Peer Review File · Nature Human Behaviour]

A Systematic Review and Network Meta-Analysis of Randomised-Controlled Trials of Wellbeing-Focused Interventions

Corresponding Author: Professor Andrew Kemp

Version 0:

Decision Letter:

15th May 2025

Dear Professor Kemp,

Thank you once again for your manuscript, entitled "A Systematic Review and Network Meta-Analysis of Randomised-Controlled Wellbeing-Focused Interventions," and for your patience during the peer review process.

Your manuscript has now been evaluated by 2 reviewers, whose comments are included at the end of this letter. Although the reviewers find your work to be of interest, they also raise some important concerns. We are [very] interested in the possibility of publishing your study in Nature Human Behaviour, but would like to consider your response to these concerns in the form of a revised manuscript before we make a decision on publication.

To guide the scope of the revisions, the editors discuss the referee reports in detail within the team, including with the chief editor, with a view to (1) identifying key priorities that should be addressed in revision and (2) overruling referee requests that are deemed beyond the scope of the current study. We hope that you will find the prioritised set of referee points to be useful when revising your study. Please do not hesitate to get in touch if you would like to discuss these issues further.

1. Reviewer 1 asked that you revise your manuscript to be more tentative in their conclusions. We ask that you follow the reviewer's advice and also provide a nuanced and careful interpretation of sensitivity analyses, especially when single studies affect the results.

2. Reviewer 2 suggests that a Component Network Meta-Analysis would have been a better approach to determine whether interventions that target multiple domains (e.g., physical activity performed in nature or combined with a psychological intervention) are more effective than those with a single focus. We leave it up to you to rerun your analysis, but encourage you to consider this alternative.

3. Please specify and clearly motivate any deviations from the protocol in the main text.

In sum, we invite you to revise your manuscript taking into account all reviewer and editor comments. We are committed to providing a fair and constructive peer-review process. Do not hesitate to contact us if there are specific requests from the reviewers that you believe are technically impossible or unlikely to yield a meaningful outcome.

We hope to receive your revised manuscript within two months. I would be grateful if you could contact us as soon as possible if you foresee difficulties with meeting this target resubmission date.

- Include a "Response to the editors and reviewers" document detailing, point-by-point, how you addressed each editor and referee comment. If no action was taken to address a point, you must provide a compelling argument. When formatting this

document, please respond to each reviewer comment individually, including the full text of the reviewer comment verbatim followed by your response to the individual point. This response will be used by the editors to evaluate your revision and sent back to the reviewers along with the revised manuscript.

- Highlight all changes made to your manuscript or provide us with a version that tracks changes.

- **EXTENDED DATA FIGURES**

Link Redacted

We look forward to seeing the revised manuscript and thank you for the opportunity to review your work. Please do not hesitate to contact me if you have any questions or would like to discuss these revisions further.

Sincerely,

[Redacted]

[Redacted]

[Redacted]

Nature Human Behaviour

Reviewer expertise:

Reviewer #1: wellbeing interventions ; RCTs

Reviewer #2: withdrawn

Reviewer #3: network meta-analysis ; meta-analysis

REVIEWER COMMENTS:

Reviewer #1 (Remarks to the Author):

Comments for the Attention of the Authors

This manuscript reports on a systematic review and network meta-analysis (NMA) investigating the comparative effectiveness of multiple wellbeing-focused interventions, including psychological interventions, physical activity, social identity building, and nature-based interventions, among the general adult population.

Strengths of the manuscript include:

- Timely and important topic that suffers from lack of conceptual clarity and limited quality research, requiring high level overviews to shape research efforts. The manuscript addresses an important research gap by integrating multiple wellbeing interventions within a single comparative framework.
- The authors clearly delineate their definition of wellbeing and thoroughly justify their choice to focus on general populations rather than clinical ones, thus enhancing the applicability of their findings.
- Systematic search strategy, rigorous data extraction, and robust sensitivity analyses.

The following issues are points the authors may wish to consider. The most substantive issues are described first, followed by additional comments as they arose in the manuscript:

1. The focus on the general population is welcome, but there is one conceptual issue that needs expansion. Models of global health such Rose include in the general population those with ill health, those with some symptoms, those who are well, and those who are thriving. That is to say the general population is dimensional and encompasses those with mental ill health and those who are languishing.
2. Inevitably any attempts at categorization of such a diverse field will involve heterogeneity. This is especially true for "nature-based" and "social identity building," where further specification of the diversity within these categories and how this influenced the network structure and interpretation of results would be useful. However systematic the methodology, this

cannot be smoothed out. I would suggest the discussion and abstract are more tentative in their conclusions. For example, in the abstract: "However, nature-based interventions were characterised by small samples and a moderate-to-high risk of bias. Multiple sub-group and sensitivity analyses confirmed suggest these interventions deserve further study but must address issues of conceptual clarity and methodological rigor.

3. Sensitivity Analyses Interpretation: The manuscript notes variability in the effectiveness of Acceptance and Commitment Therapy (ACT) across sensitivity analyses, influenced notably by a single study (Danitz, 2014). Greater clarity on how such variations might affect broader interpretations and recommendations for practice would be valuable.

4. Publication Bias and Methodological Rigor: While the manuscript acknowledges potential publication bias indicated by funnel plot asymmetry, further discussion of how future research might systematically address this bias, potentially including grey literature, would strengthen the manuscript.

5. I would suggest restricting the introduction to background directly relevant and part of the review and analysis. The overarching model seems helpful, especially if its returned to in the discussion but discussion of the vagal nerve is a distraction and not really addressed in any of the studies.

Additional Comments:

- The title is grammatically not quite right – suggest: A Systematic Review and Network Meta-Analysis of Randomised-Controlled Wellbeing -> A Systematic Review and Network Meta-Analysis of Randomised-Controlled Trials of Wellbeing-Focused Interventions."
- The manuscript occasionally lacks clarity in distinguishing between direct and indirect comparisons in the NMA; additional clarification on the implications of indirect comparisons on overall confidence in findings is recommended.
- Figure and table labelling could be improved for immediate comprehension, especially for readers less familiar with network meta-analysis methodologies.

Thank you for the opportunity to review this manuscript. I look forward to seeing the manuscript progress further.

In summary, the work clearly represents a substantial contribution to an important field, but the conclusions are in my view framed more definitively than is justified.

Reviewer #3 (Remarks to the Author):

This is a review focusing on the methodology

This manuscript presents a systematic review and meta-analysis of interventions aimed at improving well-being. The methodology is sound, and the authors have followed all necessary steps to conduct a high-quality meta-analysis. Therefore, I believe the findings successfully contribute to a better understanding of interventions aimed at increasing well-being in non-clinical populations. Still, I have a few suggestions for improvement:

-In the introduction, the authors state "We also aim to determine whether interventions which target multiple domains (e.g., physical activity performed in nature or combined with a psychological intervention) are more effective than those with a single focus". To determine this, a Component Network Meta-Analysis (CNMA) would have been a better approach. Instead of treating interventions as whole entities, CNMA analyzes the effects of individual components within interventions to identify which elements contribute most to their effectiveness. This method could help answer interesting research questions, such as the exact added value of exercise on top of psychological interventions. While I am not suggesting that the authors should rerun their analyses using this more advanced technique, it would certainly be interesting to see the results.

Rücker, G., Petropoulou, M., & Schwarzer, G. (2020). Network meta-analysis of multicomponent interventions. *Biometrical Journal*, 62(3), 808-821.

Tsokani, S., Seitidis, G., & Mavridis, D. (2023). Component network meta-analysis in a nutshell. *BMJ Evidence-Based Medicine*, 28(3), 183-186.

-Please specify if there are any deviations from the protocol.

-I wonder why the moderator analyses (subgroup analyses) were conducted in the pairwise meta-analyses rather than within the network meta-analysis. While I do not believe the results would have changed significantly, given that the manuscript focuses on network meta-analysis and that moderator analyses can be performed within this framework, I would be interested in understanding the reasoning behind this decision.

-Additional analyses: It was found that the intensity of the intervention affects the observed effect sizes. In Supplementary Material 3.3, an F statistic is reported, which is associated with a significant p-value. However, no multiple comparisons were performed among categories, making it unclear which categories differ. This multiple comparison analysis should be conducted.

-In Figure S3 from supplementary material (S Figure 3 | Network meta-analysis model results when studies containing high risk of bias are excluded.), does the "Comparisons" column in the forest plot indicate the number of direct comparisons available? Please clarify this in the figure. If possible, could this "Comparisons" column also be included in Figure 2 of the main manuscript?

-In the funnel plot (Supplementary material 6.1.), the effect sizes for "Educational program vs C" and "Exercise +

Psychological intervention vs C” are not included, or at least, these comparisons are not listed in the legend. Why?

-I have tested the code, and everything runs smoothly except for one line: in line 65, a dataset (p7) is selected that has not been created beforehand.

```
# =====  
# Analyse Network Structure  
# =====  
nc <- netconnection(treat1,treat2,studlab,data=p7)  
nc  
nc$D.matrix
```

Error in netconnection(treat1, treat2, studlab, data = p7) : object 'p7' not found

Minor comments:

-Not sure, but I think there is an error in this sentence: “However, psychological interventions comprise of only one subset of approaches to promote wellbeing and are typically studied in isolation from interventions from other disciplines.”

-Page 6, line 165, a comma is missing in the cite of Richardson (2016)

-The title of Table 2 contains an extra “v.”

-In Table 1, the label for the last category is missing (I assume it should be EXPSY).

-On page 24, line 599, I believe the p-value is incorrect. Shouldn't it be .05?

Version 1:

Decision Letter:

12th August 2025

Dear Professor Kemp,

Thank you once again for your revised manuscript, entitled "A Systematic Review and Network Meta-Analysis of Randomised-Controlled Trials of Wellbeing-Focused Interventions," and for your patience during the re-review process.

Your manuscript has now been evaluated by the same reviewers who evaluated your original manuscript. All reviewer feedback is included at the end of this letter. Although the reviewers found your manuscript to have improved during revision, they also raise some important outstanding concerns. We remain very interested in the possibility of publishing your study in Nature Human Behaviour, but would like to consider your response to these outstanding concerns in the form of a revised manuscript before we make a decision on publication.

Specifically, we ask that you address the remaining concerns raised by Reviewer 2, and include unpublished work in your search and analysis, as well as revise the subgroup analyses.

In sum, we invite you to revise your manuscript taking into account all reviewer and editor comments. We are committed to providing a fair and constructive peer-review process. Do not hesitate to contact us if there are specific requests from the reviewers that you believe are technically impossible or unlikely to yield a meaningful outcome.

We hope to receive your revised manuscript within 4-8 weeks. I would be grateful if you could contact us as soon as possible if you foresee difficulties with meeting this target resubmission date.

- Include a “Response to the editors and reviewers” document detailing, point-by-point, how you addressed each editor and referee comment. If no action was taken to address a point, you must provide a compelling argument. This response will be used by the editors and reviewers to evaluate your revision.
- Highlight all changes made to your manuscript or provide us with a version that tracks changes.

Link Redacted

We look forward to seeing the revised manuscript and thank you for the opportunity to review your work. Please do not hesitate to contact me if you have any questions or would like to discuss these revisions further.

Sincerely,

[Redacted]

[Redacted]

[Redacted]

Nature Human Behaviour

REVIEWER COMMENTS:

Reviewer #1 (Remarks to the Author):

The authors have been responsive to the feedback and in my view an already very good manuscript is improved further.

Reviewer #2 (Remarks to the Author):

I would like to thank the authors for addressing most of the concerns raised in my previous review. I have also run the R code, and it executes smoothly.

However, I still have some concerns about the moderator analyses (pages 7 and 8). First, the authors should explain how "subgroup analyses" differ from meta-regression analyses. They state that they performed both types of analyses to assess the effect of intervention intensity, but in principle, subgroup analyses and meta-regression analyses are statistically the same type of analysis. Could the authors clarify which specific type of analysis was carried out in each case? I can see from the R code that they use the `metagen()` function for subgroup analyses and the `rma()` function for meta-regression, but don't these lead to the same results, given that the underlying statistical model being implemented is the same?

Second, for these subgroup analyses, they group all active interventions into a single category. It is possible that there is an interaction between the active treatments and the moderator variable (e.g., a specific intervention may be more effective when delivered on-site rather than online). Testing the interaction between active treatment type and the moderator effect could provide additional information that would be useful to the field.

Finally, I apologize for not raising this concern in my previous review, but I believe it is a mistake not to include unpublished data. Including studies that have not been published is one of the few ways to mitigate the effect of publication bias, and unpublished studies are not necessarily of low quality. Since the authors also perform a risk-of-bias assessment and re-analyze the data excluding high-risk-of-bias studies, I see no danger (and only potential benefits) in including unpublished studies.

Version 2:

Decision Letter:

Our ref: NATHUMBEHAV-25010387B

18th September 2025

Dear Dr. Kemp,

Thank you for submitting your revised manuscript "A Systematic Review and Network Meta-Analysis of Randomised-Controlled Trials of Wellbeing-Focused Interventions" (NATHUMBEHAV-25010387B). We will be happy in principle to

publish it in Nature Human Behaviour, pending minor revisions to satisfy the referees' final requests and to comply with our editorial and formatting guidelines.

We are now performing detailed checks on your paper and will send you a checklist detailing our editorial and formatting requirements within two weeks. Please do not upload the final materials and make any revisions until you receive this additional information from us.

Sincerely,

[REDACTED]

[REDACTED]

[REDACTED]

Nature Human Behaviour

Response to Editor

We thank the editor for the opportunity to revise our manuscript and are happy to submit our updated version. This paper delivers a comprehensive comparison of wellbeing interventions, synthesising 183 RCTs in general population samples. It reinforces confidence in established psychological and mind–body approaches such as mindfulness, yoga, and compassion-based therapies and identifies emerging promise in integrated interventions combining physical activity with psychological strategies. By using network meta-analysis, the study enables cross-domain comparisons that were not previously possible, and it also exposes critical evidence gaps particularly for nature-based and social interventions and calls for a new wave of high-quality, transdisciplinary research.

We have addressed each of the prioritised points:

1. **Refined interpretation of conclusions:** In response to Reviewer 1, we refined the tone of the abstract, results, and discussion to reflect the complexity of the evidence base, without diminishing the importance or implications of our findings. Our revised language maintains the significance of the results while ensuring that conclusions are appropriately contextualised in terms of heterogeneity, strength of evidence, and methodological limitations. We have also clearly explained the particular case of Acceptance and Commitment Therapy (ACT) in which an individual study substantially influenced estimates. We have clarified the interpretation of sensitivity analyses which support our confidence in our main effect sizes and rankings.
2. **Component Network Meta-Analysis (CNMA):** we explored the feasibility of CNMA but found it unsuitable for our dataset due to sparse and inconsistently reported components in the available literature, and proceeding with this methodological approach would violate its assumptions. We have written an extensive rationale for our decision not to use CNMA in our response to reviewer 3 below. We have also added a paragraph to the Discussion acknowledging CNMA's potential for future research once further studies have been published on this topic.
3. **Protocol deviations:** We now make it clear in the Methods section that there were no deviations from the registered PROSPERO protocol regarding inclusion criteria or primary outcomes. We have further clarified that the node definitions were refined iteratively based on available data, with detailed documentation of this process provided in the Supplementary Materials.

We have also updated figure legends and tables for clarity and addressed all minor issues raised by the reviewers. We hope these revisions meet editorial expectations and thank you for considering our revised manuscript.

Response to Reviewer 1

Reviewer 1 – Comment 1:

“The focus on the general population is welcome, but there is one conceptual issue that needs expansion. Models of global health such as Rose include in the general population those with ill health, those with some symptoms, those who are well, and those who are thriving. That is to say the general population is dimensional and encompasses those with mental ill health and those who are languishing.”

Author Response:

We agree that the term “general population” should be understood as encompassing a full spectrum of mental health states from individuals who are flourishing, to those who are languishing or experiencing subclinical symptoms. Our inclusion/exclusion criteria excluded studies focused on clinical populations (i.e., individuals with a formal mental health diagnosis or undergoing medical treatment). However, we acknowledge that participants in many included studies may still experience symptoms or psychological distress.

We have revised the introduction and discussion to incorporate a dimensional framing of mental health, explicitly citing Rose's model of population health.

Revised Manuscript in Introduction:

"To enhance the scope of this research we focus on general population samples rather than specific clinical groups. We acknowledge the general population may however span individuals who are flourishing, languishing, or experiencing subclinical symptoms or psychological distress (Keyes, 2007). This supports our aim of maintaining methodological rigor, whilst increasing generalisability of findings and identifying scalable, preventative approaches with potential to positively shift the distribution of wellbeing at scale (Rose, 1985)."

Revised Manuscript in Discussion:

"A key strength of this study is its focus on participants from the general population, selected with consideration for both public health impact and methodological rigour. Excluding clinical samples helped preserve the assumption of transitivity as high clinical heterogeneity between patient groups would likely undermine this assumption by introducing systematic differences unrelated to the interventions. While our criteria excluded studies targeting individuals with diagnosed mental illness or undergoing treatment, we recognise that participants across included trials likely represented a wide spectrum of wellbeing, from flourishing to languishing, including those experiencing subclinical symptoms (Keyes, 2007). This broader inclusion enhances the generalisability of our findings. In line with Rose's population health model (1985), improvements in wellbeing, when applied at scale, can yield substantial societal benefits."

Reviewer 1 – Comment 2:

"Inevitably any attempts at categorization of such a diverse field will involve heterogeneity. This is especially true for "nature-based" and "social identity building," where further specification of the diversity within these categories and how this influenced the network structure and interpretation of results would be useful. However systematic the methodology, this cannot be smoothed out. I would suggest the discussion and abstract are more tentative in their conclusions. For example, in the abstract: "However, nature-based interventions were characterised by small samples and a moderate-to-high risk of bias. Multiple sub-group and sensitivity analyses confirmed suggest these interventions deserve further study but must address issues of conceptual clarity and methodological rigor."

Author Response:

We agree that the categories of nature-based and social identity-building interventions were conceptually broad and methodologically diverse. We have made several revisions to specify the diversity within these categories and how this influenced the network structure. Given the limited word count for the abstract, further nuance is instead added to discussion.

Revised Manuscript on Nature Interventions (Discussion)

In the discussion, we have now expanded our reflection on the nature-based interventions node. We explicitly list the types of interventions included (e.g., horticultural therapy, nature photography, outdoor walks), and discuss the conceptual and delivery-related heterogeneity that likely introduced uncertainty into the pooled estimate. We also now clarify that this node relied heavily on indirect evidence and was composed of small, often high-risk-of-bias studies. We also gave a suggestion on how to improve conceptual clarity in future using nature connectedness as a psychological mechanism targeted via the intervention compared to any mere nature exposure. The updated paragraph reads as follows:

“Contrary to expectations, nature-based interventions did not significantly outperform control conditions. However, this finding should be interpreted cautiously. The nature-based node was weakly integrated within the broader network and based largely on indirect evidence. Studies in this group were small, at moderate-to-high risk of bias, and conceptually diverse, ranging from horticultural therapy and nature photography to guided meditation in green spaces. While all took place in natural environments, they varied widely in their psychological aims, delivery formats, and therapeutic mechanisms. This conceptual heterogeneity likely diluted the pooled effect estimate and limits interpretability. To improve clarity in future research, we suggest defining nature-based interventions not merely by setting, but by whether they actively cultivate nature connectedness as a psychological mechanism. Growing evidence suggests that interventions designed to deepen emotional and sensory engagement with nature, rather than simply providing exposure are more effective in promoting wellbeing and encouraging pro-environmental behaviour (Capaldi et al., 2014; Richardson et al., 2019). Future trials and reviews should explicitly distinguish between these approaches.”

Revised Manuscript on Social Identity Interventions (Discussion)

Social identity-building interventions were initially considered for inclusion in the network. In response to the reviewer’s request, we now provide a fuller explanation of why this node was ultimately excluded, and we have added conceptual nuance to our definition.

“Our protocol defined social identity–building interventions as those aiming to form or strengthen a shared sense of group identity or belonging, grounded in social identity theory (Haslam et al., 2009). In practice, however, identifying suitable RCTs proved challenging. Few studies explicitly referenced “social identity,” and those that were potentially eligible varied widely in content including discussions of current-events (Rattenbury et al., 1989), reminiscence groups (Yousefi et al., 2015), parenting workshops (Chesak et al., 2020), and emotional peer support (Hirani et al., 2018), making it difficult to define a conceptually coherent node. Additionally, nearly all studies in this category were conducted with older adult populations, whereas other intervention types typically involved mixed-age samples. Given that NMA assumes a comparable distribution of effect modifiers across nodes, this demographic imbalance violates the assumption of transitivity. We inspected covariate distributions and determined that including this node would compromise the validity of the NMA. For these reasons, we excluded social identity–building interventions from the final network. Future research would benefit from more explicit operationalisation of social identity mechanisms and clearer reporting of group dynamics within interventions.”

Reviewer 1 - Comment 3:

“The manuscript notes variability in the effectiveness of Acceptance and Commitment Therapy (ACT) across sensitivity analyses, influenced notably by a single study (Danitz, 2014). Greater clarity on how such variations might affect broader interpretations and recommendations for practice would be valuable.”

Author response:

We have now revised the Discussion to clarify that variability in ACT’s effect size across sensitivity analyses was driven by this single study, which had methodological limitations and artificially reduced its effect size. We clarify that we have confidence in our main ACT pooled estimate because it is consistent with a large, low-risk RCT (Viskovich & Pakenham, 2020; n = 1,162), which reported an identical effect size. We caution against overinterpreting these rank shifts, due to their overlapping confidence intervals. Given that we understand the cause of this inconsistency is not because of strong evidence against ACT, we do not change our broader recommendation for practice.

Revised manuscript (Discussion):

“The variability in ACT’s effectiveness across sensitivity analyses was largely driven by one study: Danitz (2014). This high-risk-of-bias trial used only post-test scores and measured wellbeing with the Philadelphia Mindfulness Scale, which includes “awareness” and “acceptance” subscales. We extracted the “awareness” subscale, which aligned more closely with our ‘positively framed’ definition of wellbeing, but ACT only significantly improved “acceptance”, whilst baseline imbalances biased the awareness scores in favour of the control group, artificially lowering ACT’s effect. Danitz was excluded from sensitivity analyses removing high-risk which raised ACT’s effect size from SMD = 0.37 to 0.46 and improved its ranking from seventh to fourth. However, in further analysis that excluded small studies, the Danitz study remained in the analysis, suppressing the overall estimate and rendering it non-significant. Importantly however, our main effect estimate aligned with that from a large RCT of ACT associated with low risk of bias (Viskovich & Pakenham, 2020; n = 1,162; d = 0.37), and confidence intervals overlapped with those from sensitivity models. Therefore, we consider ACT’s relative effectiveness to be robust, with no meaningful impact on conclusions or implications for practice.”

Reviewer 1 - Comment 4:

“Publication Bias and Methodological Rigor: While the manuscript acknowledges potential publication bias indicated by funnel plot asymmetry, further discussion of how future research might systematically address this bias, potentially including grey literature, would strengthen the manuscript.”

Author response:

We have expanded our discussion to more clearly address how future meta-analyses might mitigate publication bias. Specifically, we highlight the potential inclusion of grey literature, direct searches of trial registries and broader improvements in trial transparency and reporting.

“Although we excluded grey literature to preserve methodological rigour, future meta-analyses could benefit from its cautious inclusion but only if rigorous quality appraisal is possible (Page et al., 2021; Higgins et al., 2022). Additional steps by researchers in the field, including trial registry searches and the greater uptake of prospective registration and outcome reporting, would help reduce the risk of publication bias and improve the completeness and transparency of the evidence base.”

Reviewer 1 - Comment 5:

“I would suggest restricting the introduction to background directly relevant and part of the review and analysis. The overarching model seems helpful, especially if it’s returned to in the discussion, but discussion of the vagal nerve is a distraction and not really addressed in any of the studies.”

Author Response:

We have removed discussion of vagal function from the introduction and have streamlined the conceptual framing to ensure all content directly supports the aims and scope of the current review.

Additional comments

Reviewer comment: “The title is grammatically not quite right – suggest: A Systematic Review and Network Meta-Analysis of Randomised-Controlled Wellbeing -> A Systematic Review and Network Meta-Analysis of Randomised-Controlled Trials of Wellbeing-Focused Interventions.”

Author Response: We have revised the title in line with your suggestion.

Reviewer comment: *“The manuscript occasionally lacks clarity in distinguishing between direct and indirect comparisons in the NMA; additional clarification on the implications of indirect comparisons on overall confidence in findings is recommended.”*

Author Response: We thank the reviewer for highlighting the need to clarify how direct and indirect comparisons contribute to the interpretation of our network meta-analysis. To address this, we added a paragraph to the discussion section that explains the concept of indirect evidence and how we tested for consistency (i.e., whether direct and indirect comparisons yield similar results). We also explicitly noted that no significant inconsistency was found, thereby strengthening confidence in our pooled effect estimates (see page 13 in discussion).

Revised Manuscript, Discussion

“Because few interventions had been tested head-to-head, many comparisons in our network relied on indirect evidence. This is a key strength of NMA, it enables comparison across interventions even when direct trials are lacking. To ensure the validity of these indirect estimates, we formally assessed consistency (that indirect and direct comparisons yield similar results) and found no significant inconsistency across the network (see Supplementary Section 4.1). This agreement between direct and indirect evidence increases confidence in the reliability of our pooled effect estimates.”

We have also clarified in the methods that an NMA estimate is a pooled estimate using both direct and indirect evidence.

Revised Manuscript, Methods

“The network meta-analysis combined both direct (head-to-head) and indirect comparisons across trials to generate pooled estimates of standardised mean differences (SMDs) between interventions. This approach enables the comparison of multiple interventions simultaneously, even when few direct comparisons exist, and strengthens the precision of effect estimates by incorporating all available evidence.”

Reviewer comment: *“Figure and table labelling could be improved for immediate comprehension, especially for readers less familiar with network meta-analysis methodologies.”*

Author Response: We have revised all figure and table captions for clarity and accessibility, including clearer descriptions of what is displayed and how to interpret key metrics such as effect sizes and P-scores. This has also been completed in supplementary materials.

“Figure 1. Network plot of intervention comparisons.

Node size reflects the number of participants; edge thickness indicates the number of direct trials comparing the two connected interventions. "C" refers to control arms.

Figure 2. Forest plot of NMA pooled estimated effect sizes for each intervention (NMA primary model). Standardised mean differences (SMDs) with 95% confidence intervals are shown. Interventions are ranked by P-score (higher = more effective).

Figure 3. Intervention ranking based on P-scores.

Bar lengths and colours indicate relative intervention ranking from the NMA (higher P-score = higher ranking).”

Final Remarks to Reviewer 1

In response to your feedback, we have revised the manuscript to adopt a more nuanced and balanced tone. We have clarified areas of conceptual heterogeneity, and more precisely contextualised the

sensitivity analyses, indicating where findings are robust and where caution is warranted. The revisions strengthen the manuscript's scientific credibility while preserving its central message: that across the 183 RCTs, we identified consistent evidence for the effectiveness of diverse wellbeing interventions, many of which showed comparable effect sizes despite originating from distinct disciplinary traditions. This represents the first quantitative synthesis to enable cross-domain comparisons and highlights the importance of taking body-based and integrative approaches as seriously as traditional psychological interventions in efforts to improve population wellbeing. Thank you for your helpful feedback and enabling us to strengthen our manuscript.

Response to Reviewer 3

Reviewer 3 - Comment 1: Suggestion of Component Network Meta-Analysis (CNMA)

"In the introduction, the authors state "We also aim to determine whether interventions which target multiple domains (e.g., physical activity performed in nature or combined with a psychological intervention) are more effective than those with a single focus". To determine this, a Component Network Meta-Analysis (CNMA) would have been a better approach. Instead of treating interventions as whole entities, CNMA analyzes the effects of individual components within interventions to identify which elements contribute most to their effectiveness. This method could help answer interesting research questions, such as the exact added value of exercise on top of psychological interventions. While I am not suggesting that the authors should rerun their analyses using this more advanced technique, it would certainly be interesting to see the results."

Author Response:

We thank the reviewer for raising the potential of using Component Network Meta-Analysis (CNMA). We were, in fact, originally open to this analytical approach and explored its feasibility during the early planning stages of this study. However, we concluded that CNMA was not feasible for several key reasons including a lack of overlapping components across studies, sparse data on cross-domain interventions, poorly defined and inconsistently reported components.

Firstly, while CNMA is well-suited to decomposing interventions into their active ingredients, it relies on sufficient representation and overlap of components across trials. In our dataset, many interventions were delivered as standalone interventions (e.g., MBSR, ACT, Exercise), with limited component overlap. Others were multi-component but lacked transparent reporting about the specific content, making reliable component coding unfeasible.

Secondly, whilst we were theoretically motivated to examine interventions that extend beyond psychological components (e.g., incorporating mind–body or environmental dimensions of wellbeing), very few trials tested such combinations. The "Exercise + Psychological" (EXPSY) node included only three studies that met our inclusion criteria. Similarly, we initially attempted to construct an "Exercise in Nature" (EXNAT) node, but only three highly heterogeneous studies were identified (e.g., a remote camping programme, a park prescription with counselling, and a brief nature video). This lack of consistent, detailed reporting reflects a broader limitation in the evidence base and highlights the need for improved intervention transparency in future trials (Michie et al., 2009; Hoffmann et al., 2014).

Third, we also considered component-level distinctions within psychological approaches. For example, we initially coded Positive Psychology Interventions (PPIs) by subtype such as gratitude, best possible self, or strengths-based exercises. However, the number of studies per component was insufficient for valid statistical synthesis. Even in multi-component PPI, the combinations of PPI subtypes varied widely. Attempting coding a CNMA in this context would have resulted in a highly fragmented network (many different PP components) with low statistical power and unreliable estimates. Where possible, we

attempted to evaluate multi-component effects through designated nodes (e.g., MPPI, EXPSY). In the meantime, we believe that our use of standard NMA strikes a pragmatic balance, allowing us to make the best use of the current evidence.

This experience reinforces one of our key conclusions: future research should focus on well-powered, theory-driven trials that test clearly specified combinations of components, particularly those integrating physical, psychological, and environmental factors. Such developments would make more sophisticated meta-analytic approaches like CNMA more viable in the future.

Revised Manuscript, Discussion:

“We initially considered a Component Network Meta-Analysis (CNMA) to examine the additive effects of intervention components (e.g., mindfulness, physical activity) (Rücker et al., 2020; Tsokani et al., 2023). However, this approach was not viable due to limited component overlap, poor reporting of intervention content, and a lack of multi-component designs. For example, only three studies met criteria for the “Exercise + Psychological” node, each with differing content. These limitations led us to adopt a standard NMA with designated nodes for multi-component interventions where feasible. Future studies which explore multi-component and cross-discipline interventions with clearer reporting could support CNMA and help identify active ingredients across intervention types.”

Reviewer 3 - Comment 2: *“Please specify if there are any deviations from the protocol.”*

Author Response:

Revised manuscript (Methods section – Protocol and Registration):

“This review was pre-registered on PROSPERO (CRD42023403480), where the study design, inclusion criteria, outcome measures, and use of network meta-analysis were pre-specified. No deviations from the registered protocol occurred. The development and refinement of intervention nodes based on the data extracted are fully documented in Supplementary Materials 2.”

Reviewer 3 - Comment 3:

“I wonder why the moderator analyses (subgroup analyses) were conducted in the pairwise meta-analyses rather than within the network meta-analysis. While I do not believe the results would have changed significantly, given that the manuscript focuses on network meta-analysis and that moderator analyses can be performed within this framework, I would be interested in understanding the reasoning behind this decision.”

Author Response

We conducted moderator analysis using pairwise meta-analysis for at least two reasons. First, we were unable to use continuous covariates as effect modifiers using netmeta R package as network meta-regression requires a more advanced (Bayesian) model, and was therefore beyond the scope of the current synthesis. Second, our moderators were categorical, which would have required stratifying multiple NMAs (i.e., up to four NMAs per moderator), leading to sparse networks.

In contrast, the pairwise approach (comparing all active interventions vs. control) allowed us to:

- Collapse across treatment types to improve power,
- Avoid assumptions of network connectivity

- Examine broad moderator effects (e.g., format, delivery, intensity) across the entire set of active interventions.

Importantly, we treat these analyses as exploratory and report them with appropriate caution. To complement this approach, we also conducted three sensitivity analyses including studies associated with low risk of bias, larger studies ($N < 45$, based on the lower quartile) and subjective wellbeing measures using full NMA models, and placed greater emphasis on those findings that were robust and able to be interpreted within the network framework.

Changes to Manuscript:

We have clarified this rationale in the Methods section (Moderator Analyses).

Revised Manuscript: – Methods Section

“We conducted six exploratory subgroup analyses to assess whether the pooled effect of all active interventions versus control varied across pre-specified effect modifiers. These were conducted using pairwise meta-analyses, which offered greater statistical power. Moderator variables included: (1) mode of delivery (e.g., in-person, online platform, live video conferencing, or self-guided); (2) treatment format (individual vs. group); (3) intervention intensity (total weeks); (4) setting (e.g., university, workplace, community, online); (5) country (Western vs. non-Western); and (6) participant age (younger vs. older than the mean sample average of 38 years). This approach allowed us to explore broad moderation patterns without fragmenting the network.”

Reviewer 3 - Comment 4:

“Additional analyses: It was found that the intensity of the intervention affects the observed effect sizes. In Supplementary Material 3.3, an F statistic is reported, which is associated with a significant p-value. However, no multiple comparisons were performed among categories, making it unclear which categories differ. This multiple comparison analysis should be conducted.”

Author Response:

Thank you for raising this. As requested, we conducted post-hoc multiple comparisons among intensity categories which indicated a significant difference between short and medium-length interventions. We also performed a mixed-effects meta-regression including all studies with intensity data, which supported intensity as an overall moderator, though no individual contrasts reached significance. We have reported both analyses in the manuscript (Results and Supplementary Materials) and clarified in the Discussion that these results should be considered exploratory. See full explanation and results in supplementary materials 3.3.1.

Revised Manuscript: (Results: Additional Analyses)

“Intervention intensity (brief, short, medium, or long) significantly moderated effects ($p = 0.001$), with medium-length interventions (5–8 weeks) showing the largest effect size (SMD = 0.57, 95% CI [0.45, 0.69], $I^2 = 81.9\%$). Post-hoc comparisons revealed that medium-length interventions significantly outperformed short interventions ($p < 0.001$). Meta-regression confirmed intensity as a significant moderator overall ($F(3,167) = 4.38$, $p = 0.0054$), though it did not identify significant differences between individual categories. Full results are provided in Supplementary Information 3.”

Revised Manuscript: (Discussion)

“Our exploratory analyses suggest that intervention intensity may meaningfully influence wellbeing outcomes. Medium-length (5–8 week) and brief interventions were associated with the largest pooled effect sizes. Post-hoc comparisons confirmed that medium interventions significantly outperformed short ones,

and meta-regression supported intensity as a moderator overall, though it did not detect statistically significant pairwise contrasts. This pattern may reflect a non-linear relationship in which very brief interventions can be impactful if focused and engaging, while longer programmes may suffer from participant burden or reduced adherence. High within-group heterogeneity and limited power may have further obscured differences across some categories. These findings highlight the potential benefit of medium-length interventions but underscore the need for well-powered trials directly comparing different intervention durations to establish optimal program length for enhancing wellbeing.”

Reviewer 3 - Comment 5:

“In Figure S3 from supplementary material (S Figure 3 | Network meta-analysis model results when studies containing high risk of bias are excluded.), does the “Comparisons” column in the forest plot indicate the number of direct comparisons available? Please clarify this in the figure. If possible, could this “Comparisons” column also be included in Figure 2 of the main manuscript?”

Author Response:

The “Comparisons” (k) column indicates the number of studies providing direct comparisons between each intervention and the reference group (Control). Estimates based only on indirect evidence may appear even when k=0. We have clarified this in the updated figure legend.

Due to journal guidelines restricting figure size and content, we initially omitted the k column from Figure 2 in the main manuscript. However, we have now included it in the revised Figure 2 to address the reviewer’s comment.

Reviewer 3 - Comment 6:

“In the funnel plot (Supplementary material 6.1.), the effect sizes for “Educational program vs C” and “Exercise + Psychological intervention vs C” are not included, or at least, these comparisons are not listed in the legend. Why?”

Author Response:

Interventions “Educational program vs Control” and “Exercise + Psychological intervention vs Control” had only a single direct comparison each. Consequently, they did not appear in the original funnel plot because this version was designed to evaluate asymmetry across multiple studies per comparison.

To address this we recalculated the funnel plot using only *direct comparisons* of all active interventions versus control, rather than the original plot, which was centered on comparison-specific pooled estimates. This updated approach aligns more closely with standard methods for assessing publication bias in meta-analysis, based on direct published evidence. The revised funnel plot is now included in the supplementary material (see updated SFigure 6). Conclusions and results of Egger's test are not changed.

Reviewer 3 - Comment 7:

“I have tested the code, and everything runs smoothly except for one line: in line 65, a dataset (p7) is selected that has not been created beforehand.”

Author Response:

This was a typo error and the data label should have been ‘p’ not ‘p7’. This has now been corrected in the R script. I have tested and can confirm it now runs correctly.

Reviewer 3: Minor Comments

1. "Not sure, but I think there is an error in this sentence: "However, psychological interventions comprise of only one subset of approaches to promote wellbeing and are typically studied in isolation from interventions from other disciplines."
2. "Page 6, line 165, a comma is missing in the cite of Richardson (2016)"
3. "The title of Table 2 contains an extra "v."
4. "In Table 1, the label for the last category is missing (I assume it should be EXPSY)"

Author response

All above minor comments were rectified as suggested.

5. "On page 24, line 599, I believe the p-value is incorrect. Shouldn't it be .05?"

Author response

We have changed to the correct value of 0.1. A p-value threshold of <0.1 was chosen for SIDE tests, as a higher threshold can increase sensitivity to potential inconsistency. This conservative choice reduces the risk of overlooking inconsistency by flagging even moderate evidence ($p < 0.1$) as potentially inconsistent.

Final Remarks to Reviewer 3

We really appreciate your thoughtful feedback and expertise in NMA. As noted, we explored CNMA, and unfortunately found it impractical in this instance given our data, and we've highlighted it as a promising direction for future research. We clarified that there were no protocol deviations and added the post-hoc comparisons you requested. We also updated figures, funnel plots, and fixed coding issues, incorporating all your helpful editorial suggestions. Thank you again for your valuable input in strengthening our manuscript.

References

- Hoffmann, T. C., et al. (2014). Better reporting of interventions: template for intervention description and replication (TIDieR) checklist and guide. *BMJ*, 348, g1687. <https://doi.org/10.1136/bmj.g1687>
- Michie, S., Fixsen, D., Grimshaw, J. M., & Eccles, M. P. (2009). Specifying and reporting complex behaviour change interventions: the need for a scientific method. *Implementation Science*, 4(1), 40. <https://doi.org/10.1186/1748-5908-4-40>

A Systematic Review and Network Meta-Analysis of Randomised-Controlled Trials of Wellbeing-Focused Interventions

Wilkie et al

Reviewer Comment 1

First, the authors should explain how “subgroup analyses” differ from meta-regression analyses. They state that they performed both types of analyses to assess the effect of intervention intensity, but in principle, subgroup analyses and meta-regression analyses are statistically the same type of analysis. Could the authors clarify which specific type of analysis was carried out in each case? I can see from the R code that they use the metagen() function for subgroup analyses and the rma() function for meta-regression, but don't these lead to the same results, given that the underlying statistical model being implemented is the same?

Author Response 1

In the original submission, our analyses of intervention intensity used two approaches:

1. Subgroup analyses (metagen function): These pooled studies within predefined intensity categories (e.g., brief, short, medium, long) and compared subgroup-specific random-effects estimates. This approach focuses on differences in the *pooled effects* between categories.
2. Meta-regression (rma function): Since intervention intensity was the only category which was significant, intervention intensity was entered as a categorical moderator at the *study level*. This allowed us to formally test whether variation in effect sizes across studies was explained by intensity, while also accounting for within-subgroup heterogeneity.

Upon reflection, and based on the reviewer's comment, we agree that presenting both approaches is redundant. To streamline the manuscript and ensure consistency, we have now re-run all moderator analyses using meta-regression models only. These results are indeed the same as the original subgroup analyses (i.e., intervention intensity was the only significant moderator, with the same pattern of effects). We believe this revision improves clarity for the reader and addresses the reviewer's concern. Please see supplementary materials 3 for full results. R script and manuscript have been updated accordingly.

Reviewer Comment 2

Second, for these subgroup analyses, they group all active interventions into a single category. It is possible that there is an interaction between the active treatments and the moderator variable (e.g., a specific intervention may be more effective when delivered on-site rather than online). Testing the interaction between active treatment type and the moderator effect could provide additional information that would be useful to the field.

Author Response 2

To address the reviewer's comment, we ran exploratory interaction analysis using meta-regression (i.e., interventions x moderator). To assess feasibility, we first cross-tabulated treatment x moderator cells and restricted analyses to intervention types with sufficient data (Mindfulness, Exercise, Combined Psychological, Single PPI). We then fit interaction meta-regression models for delivery mode, format, country (Western vs. non-Western), and intervention length. Three of these models (delivery, format, country) showed no evidence of interaction.

The length x treatment model reached statistical significance overall ($F(12,110) = 2.27, p = .013$), accounting for 12% of heterogeneity, but most contrasts were non-significant. The only significant effect indicated that medium-length exercise-based interventions were less effective than short exercise-based interventions. Given the sparsity of several cells and risk of type I error, we judged this exploratory result insufficiently robust for emphasis in the main text, but we have reported it transparently in the Supplementary Materials 3.2.

We agree that this is an important question for the field and should be revisited as a future research direction when more RCTs are available for all interventions.

See updated results of MS below.

“Additional analyses: Meta-Regression

We examined whether study or intervention characteristics moderated wellbeing outcomes using univariate mixed-effect meta-regression analysis. For intervention intensity, medium-length interventions (5–8 weeks) produced significantly stronger effects than short interventions (2–4 weeks; $SMD = -0.35, 95\% CI [-0.56, -0.15], p < .001$; overall $F(3,167) = 4.38, p = .005$). Brief (<2 weeks) and long (>8 weeks) interventions did not differ significantly. No evidence of moderation was found for delivery mode (in-person, online platform, self-guided, or live video), format (individual vs. group), study setting (community, university, workplace, online), country (Western vs. non-Western) or mean participant age. We also found no significant effect for wellbeing outcome measures used (subjective wellbeing, resilience, mindfulness, or positive affect), waitlist control (waitlist v no intervention control), or risk of bias (low, medium, or high). Full results are provided in Supplementary Information 3.

To examine whether moderator effects differed across intervention types, we conducted exploratory intervention x moderator interaction models that were restricted to four well-represented interventions (Mindfulness, Exercise, Combined Psychological Interventions, Single PPIs). No significant interactions were detected for delivery mode, format, or Western versus non-Western settings. For intervention intensity, the overall interaction model reached significance, but only one contrast (medium-length Exercise vs. short Exercise) was statistically significant. Given sparse data and multiple testing, this isolated finding should be considered exploratory and not interpreted as altering the main conclusions.”

Reviewer Comment 3

Finally, I apologize for not raising this concern in my previous review, but I believe it is a mistake not to include unpublished data. Including studies that have not been published is one of the few ways to mitigate the effect of publication bias, and unpublished studies are not necessarily of low quality. Since the authors also perform a risk-of-bias assessment and re-analyze the data excluding high-risk-of-bias studies, I see no danger (and only potential benefits) in including unpublished studies.

Author Response 3

In response to the reviewer’s feedback, we conducted additional searches to identify grey literature for inclusion. We conducted a targeted search of clinical trial registries, theses, dissertations, and institutional repositories and identified seven eligible, unpublished RCTs ($n = 709$ participants). We incorporated these studies into a sensitivity analysis using the network meta-analysis approach. Results were highly consistent with the primary NMA: effect size estimates shifted only minimally (≤ 0.03), no interventions changed statistical significance, there was minimal change in ranking order between mindfulness, compassion and exercise, but given their substantially overlapping confidence intervals and similar effect size estimates this

finding does not alter any interpretations or conclusions drawn from the data. This analysis helps to counter any remaining concerns relating to publication bias and adds to the robustness of our conclusions.

The following section has been added to supplementary materials 4.2.4.

“4.2.4. Network meta-analysis model including grey literature

In response to reviewer feedback, additional searches were conducted (August 2025) to identify studies in the grey literature for inclusion. We searched clinicaltrials.gov, National Institute for Health and Care Research database (includes funded, but not always published trials), ProQuest dissertations and also screened reference lists of relevant reviews.

Seven additional trials identified from grey literature searches were included (Bull-Beddows, 2020; Linford, 2020; Pitman, 2022; Prasek, 2015; Prudenzi, 2021; Salmon, 2004; Leininger, 2021). These studies added 709 participants across mindfulness (n = 213 participants), compassion (n = 103), exercise (n = 26), acceptance and commitment therapy (n = 52), and education interventions (n = 90).

The network remained well connected, with all 12 intervention nodes retained and 156 pairwise comparisons. The design-by-treatment interaction model suggested global inconsistency ($\tau^2 = 0.096$; $\tau = 0.310$; $I^2 = 75.9\%$ [72.1%; 79.4%]), but under the full design-by-treatment random effects model, Q decreased and between-design inconsistency was not significant (Q = 11.28, p = 0.891).

Across interventions, SMDs shifted only marginally (≤ 0.03) and 95% confidence intervals overlapped substantially with those in the main analysis. Treatment rankings were minimally affected: for example, compassion rose slightly above mindfulness, but both interventions remained within a similar band of effect size with overlapping CIs (SMD ≈ 0.43 – 0.48).

Overall, the inclusion of grey literature did not change the pattern of results or substantive conclusions. These findings suggest that our primary NMA results are robust to the inclusion of unpublished trials.”

SFigure 6. Sensitivity Analysis: Forest Plot of Network Meta-Analysis Including Grey Literature Studies.

The following text has been added to the manuscript

“In addition, we conducted a sensitivity analysis including seven unpublished trials identified through targeted grey literature searches (one unpublished clinical trial and six dissertations/theses), adding a further $n = 709$ participants across mindfulness, compassion, exercise, and ACT interventions (see supplementary materials 4.2.4). These studies met all other eligibility criteria. Results were highly consistent with the primary NMA: effect size estimates shifted only minimally (≤ 0.03), no interventions changed statistical significance, and only minor shifts were observed in ranking.”